# ANNEALING BRIDGES OFFLINE AND ONLINE RL

## ABSTRACT

Adopting the pretrain-finetune paradigm, offline-to-online reinforcement learning (RL) first pretrains an agent on historical offline data and then finetunes it through online interactions, aiming to leverage prior knowledge while adapting efficiently and safely to the new environment. A central challenge, however, is the tradeoff between catastrophic failure, i.e., a sharp early collapse in performance when the agent first transitions from offline to online, and the asymptotic success rate, i.e., the long-term performance the agent ultimately achieves after sufficient training. In this article, we first conduct a systematic study using various control benchmarks and find that existing offline and offline-to-online RL methods fail to simultaneously prevent catastrophic failure and achieve high asymptotic success rates. Next, we examine how offline data and conservative regularization influence this tradeoff. Then, we identify spurious Q-optimism as the key driver of collapse, i.e., early in fine-tuning, the learned value function can mistakenly rank inferior actions above those from offline training, steering the policy toward failure. Finally, we introduce Smooth Offline-to-Online Annealing for RL (SOAR), a simple but effective dual annealing scheme that gradually reduces reliance on offline data and conservative penalties, thereby mitigating catastrophic failure while improving long-term performance. We carry out extensive numerical experiments to confirm the efficacy and robustness of SOAR across diverse RL tasks.

## 1 INTRODUCTION

The pretrain-finetune paradigm has driven much of the recent success in modern machine learning across diverse domains such as natural language processing and computer vision (Min et al., 2023; Khan et al., 2024). Inspired by these advances, reinforcement learning (RL) has adopted a similar paradigm: an agent is first pretrained offline using historical static data, is then subsequently finetuned via online interactions with the target environment (Agarwal et al., 2022; Luo et al., 2024). This setting, known as offline-to-online RL, allows the agent to leverage effectively the prior knowledge, while adapting efficiently and safely to the new environment, reducing the reliance on costly and risky online interactions.

Previous studies along this direction have shown that modifying the training objective or algorithmic structure during the online phase leads to a better long-term performance than naively deploying offline RL algorithms without modification (Nakamoto et al., 2023; Zhou et al., 2025; Xiao et al., 2025). However, catastrophic failure often occurs, i.e., there is a sharp early collapse in performance when the agent first transitions from offline to online. This raises a challenging and open question: *how can we achieve both high asymptotic success rates and reduced catastrophic failure during the transition to the online phase?*

This question is particularly critical in safety-sensitive applications such as healthcare and robotics (Singh et al., 2022; Liu et al., 2020). For instance, in the medical domain, offline datasets may capture prior treatment information, while online fine-tuning involves real-time patient interactions. Catastrophic failure in this scenario could result in harmful interventions. Ideally, the agent should remain aligned with the optimal actions learned from the offline data, while using online explorations to uncover additional optimal actions not represented in the offline data.

To address this question, we first conduct a systematic study on complex and realistic robot control environments from D4RL (Fu et al., 2020), Adroit, FrankaKitchen, and AntMaze. We evaluate two representative offline RL algorithms, IQL (Kostrikov et al., 2022) and CQL (Kumar et al., 2020), as

well as three offline-to-online RL algorithms, Cal-QL (Nakamoto et al., 2023), PORL (Xiao et al., 2025), and recent state-of-the-art WSRL (Zhou et al., 2025). We analyze their catastrophic failure modes and success rates during online fine-tuning. On AntMaze, we further extend our study to the challenging ultra-diverse variant, where none of the baseline methods achieves a perfect success rate, thereby stress-testing the algorithms in the most difficult settings. We have found that the existing methods *cannot* simultaneously prevent catastrophic failure and achieve high long-term success rate.

Next, to better understand the drivers of collapse and the ingredients for high asymptotic performance, we perform a controlled study with CQL. We have found that, eliminating the conservative regularizer improves the asymptotic performance, whereas phasing out the offline data yields faster convergence. Nevertheless, removing either the conservative regularizer, which serve as inductive biases tailored to offline training, or the offline data, substantially increases the incidence of catastrophic failure. Nevertheless, removing either the conservative regularizer, an inductive bias tailored to offline training, or the offline data substantially increases the incidence of catastrophic failure.

We further hypothesize that a main driving factor for catastrophic failure is *Spurious Q-Optimism*. That is, early in online fine-tuning, the agent incorrectly reverses the relative value ordering between the actions from offline pre-training and those proposed by the current policy for the same state, causing the agent to favor the actions that later prove inferior under convergence. We quantify this effect via a new metric called *Spurious Q-Optimism Ratio (SQOR)*, which is defined as the fraction of states whose current versus final value ordering disagrees, and show that SQOR closely tracks the incidence of collapses across tasks and settings. Furthermore, we show that alternative strategies proposed in prior works (Fujimoto & Gu, 2021; Zhou et al., 2025; Xiao et al., 2025), including regularizing critic KL divergence, adjusting update-to-data (UTD) ratios, modifying warmup lengths, or tuning hyperparameters such as batch size, network dimensions and learning rates, are insufficient to effectively mitigate catastrophic failure.

Finally, motivated by these insights, we propose *Smooth Offline-to-Online Annealing for RL* (SOAR), a simple yet effective method that gradually decreases both the offline data ratio and the conservative regularizer weight $\alpha$ via annealing during online fine-tuning. Empirical results show that this dual annealing strategy lowers the incidence of catastrophic failure compared with existing baselines, while achieving superior long-term performance. We also conduct extensive ablation studies on SOAR's hyperparameters and on the contribution of each annealing component. These studies offer an actionable guidance: to prioritize stability and suppress early catastrophic failures, one may apply a single-component annealing in a task-dependent manner. We also outline concrete design choices and practical heuristics for hyperparameter selection.

Our contributions are four-fold. First, we provide a systematic study demonstrating that prevailing offline and offline-to-online methods fail to balance catastrophic failure suppression with high asymptotic success. Second, we show how offline data and conservative regularization shape this trade-off, which points to a pathway toward achieving both goals simultaneously. Third, we identify the key driving factor behind catastrophic failure and derive a metric to quantify it. Finally, we introduce SOAR, a simple yet effective dual annealing scheme that consistently reduces catastrophic failure and improves long-term performance.

## 2 RELATED WORKS

**Offline-to-Online RL.** While offline RL methods such as CQL (Kumar et al., 2020), IQL (Kostrikov et al., 2022), and others (Kostrikov et al., 2021; Tarasov et al., 2023) can be deployed online, strong online performance typically requires additional fine-tuning. Simply fine-tuning the offline objective without modification often limits gains (Nakamoto et al., 2023), motivating methods that explicitly leverage online interaction. Proposed approaches include relaxing excessive conservatism in value estimates (Nakamoto et al., 2023; Luo et al., 2024; Hu et al., 2024), inserting an adaptation phase between offline pre-training and online fine-tuning (Zhou et al., 2025; Shin et al., 2025; Xiao et al., 2025), using multiple Q-functions (Lee et al., 2022; Zhao et al., 2023), tuning the UTD ratio (Feng et al., 2024; Xiao et al., 2025), and incorporating uncertainty (Guo et al., 2023; Wen et al., 2024b). In contrast to methods that update both value functions and policies (including ours), some approaches rely exclusively on pretrained policies (Uchendu et al., 2023; Xiao et al., 2025; Hu et al., 2023).

Several works observe catastrophic failure during the transition from offline pre-training to online fine-tuning, attributing it to distributional shift and unstable Q-learning (Wen et al., 2024a;

Nakamoto et al., 2023). Many proposed remedies introduce additional computation, e.g., uncertainty estimation (Wen et al., 2024a), calibration penalties (Nakamoto et al., 2023), or actor-critic alignment (Yu & Zhang, 2023). In contrast, our method employs a minimal design based on annealing, which adds essentially no computational overhead while also improving asymptotic performance.

# 3 EXPERIMENTAL SETUP

**Environments and Datasets.** Following the evaluation protocol of WSRL (Zhou et al., 2025), we assess our method on three challenging, realistic environments: FrankaKitchen and AntMaze from D4RL (Fu et al., 2020), and the dexterous manipulation environment Adroit from AWAC (Nair et al., 2020). Within these environments, we evaluate the following tasks: for Adroit, pen-binary and door-binary; for FrankaKitchen, kitchen-mixed and kitchen-partial; and for AntMaze, antmaze-large-diverse and antmaze-large-play. All offline pre-training datasets match those used in WSRL. Further details on the tasks and datasets are provided in Appendix G.

**Training Procedure.** For training, we pretrain for 1M steps in AntMaze, 250K steps in FrankaKitchen, and 40K steps in Adroit, followed by 400K online fine-tuning steps for all tasks. Compared to WSRL (Zhou et al., 2025), which used only 300K fine-tuning steps, we extend fine-tuning steps to 400K steps to better observe asymptotic performance trends. On Adroit, we found that increasing pen-binary pre-training from 20K (used in WSRL) to 40K yields more consistent gains; for door-binary, offline pre-training variance is higher and the difference between 20K and 40K is less pronounced, but we adopt 40K to stabilize trends.

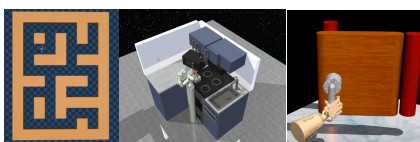

Figure 1: **Evaluated Environments.** Illustration of the environments used in our experiments: AntMaze, FrankaKitchen, and Adroit.

**Baseline Methods.** We include WSRL (Zhou et al., 2025) and PORL (Xiao et al., 2025) as recent offline-to-online methods, Cal-QL (Nakamoto et al., 2023) due to its explicit treatment of catastrophic failure, and CQL (Kumar et al., 2020), IQL (Kostrikov et al., 2022), and SAC (Haarnoja et al., 2018) to align with prior evaluations in WSRL and ensure comprehensive comparisons. Further details on all baselines are provided in Appendix F.

We largely follow the experimental setup of WSRL (Zhou et al., 2025). Modifications on the experimental setup to stabilize training are detailed in Appendix G.3. Due to computational constraints, we use five random seeds in all experiments (unless otherwise noted) and report 95% confidence intervals with shaded regions in the plots. Across all experimental results, Step 0 marks the beginning of the online fine-tuning phase.

**CQL.** We adopt CQL as the backbone of our method, following WSRL and Cal-QL. In offline RL, the agent is trained using a fixed dataset $\mathcal{D} = \{(s_i, a_i, r_i, s_i')\}_{i=1}^N$ collected by some behavior policy, without interacting with the environment. A key challenge is that standard Q-learning objectives can assign erroneously high values to actions not present in the dataset, leading to poor policy performance when deployed online. CQL (Kumar et al., 2020) addresses this by adding a regularization term to the standard Bellman error that penalizes Q-values of actions sampled from the policy relative to those from the dataset:

$$\mathcal{L}_{\text{CQL}} = \mathcal{L}_{\text{TD}} + \alpha\Big(\mathbb{E}_{s\sim\mathcal{D}, a\sim\pi}[Q_\theta(s,a)] - \mathbb{E}_{s,a\sim\mathcal{D}}[Q_\theta(s,a)]\Big), \tag{1}$$

where $\alpha > 0$ controls penalty strength. This discourages high Q-values for OOD actions.

**Metrics.** We define *catastrophic failure* as the drop between the success rate at the start of fine-tuning and the minimum success rate observed within the first 100K steps, isolating the effect of the offline-to-online transition from training stochasticity. As reported in Appendix K, extending the window to the full 400K steps yields no statistically significant change in the measured failure magnitude, which justifies our choice of a 100K step window.

We define the *asymptotic success rate* as the mean success rate over 350K-400K steps. As evident in Figure 2, all baselines have converged by 350K, and Appendix K confirms no statistically significant difference between performance at 350K and 400K. Hence, performance in this interval is a valid proxy for asymptotic behavior.

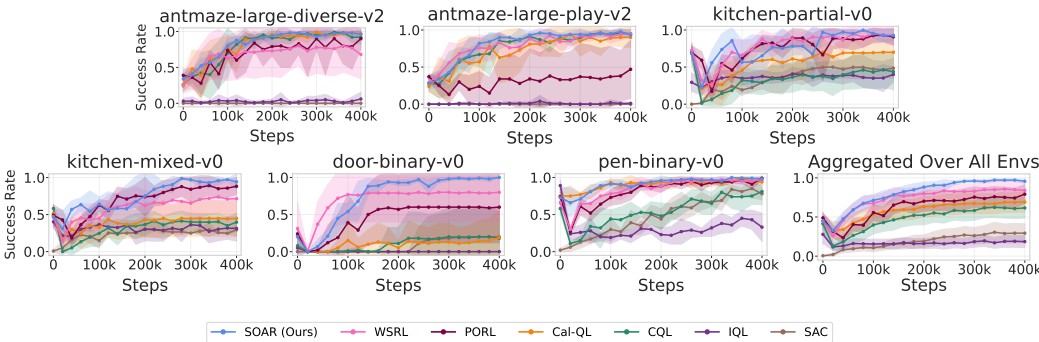

Figure 2: **Annealing Bridges Offline and Online RL.** Existing offline and offline-to-online methods fail to simultaneously mitigate catastrophic failure and attain high asymptotic success. In contrast, our method (SOAR) reduces early performance collapse and achieves superior final performance across tasks. Step 0 marks the start of online fine-tuning.

## 4 CHALLENGES IN BRIDGING OFFLINE AND ONLINE RL: PERFORMANCE VS. STABILITY

We first show that existing offline RL algorithms and offline-to-online approaches are unable to simultaneously prevent catastrophic failure during online fine-tuning and attain high asymptotic success rates. As illustrated in Figure 2, when baselines are finetuned without retaining offline data, none of the methods meet both objectives. Notably, the only objective-level difference between the offline algorithm CQL and the online algorithm SAC is CQL's conservative regularizer (Equation 1). Among offline-to-online methods, WSRL and PORL set CQL's conservative weight $\alpha$ to zero during fine-tuning, whereas CQL and Cal-QL keep $\alpha$ equal to its offline pre-training value. Empirically, WSRL and PORL achieve higher asymptotic performance than CQL and Cal-QL, but suffer larger catastrophic failures. This motivates a controlled analysis of how removing the conservative regularizer affects both outcomes. In addition, because the availability of offline data is a key distinction between the offline and online phases, we also study how retaining versus discarding offline data influences performance and stability.

## 5 ROLE OF OFFLINE DATA AND CONSERVATIVE REGULARIZATION IN ONLINE FINE-TUNING

To disentangle the effects of offline data and conservative regularization, we conduct controlled studies with CQL during online fine-tuning. As shown later, removing either $\alpha$ or offline data increases the incidence of catastrophic failure. Accordingly, when analyzing catastrophic failure, we vary one factor while holding the other fixed: we keep $\alpha$ at its offline-pre-training value when assessing the effect of offline data, and we fix the offline replay mixture at $25\%$ per update when assessing the effect of $\alpha$. Conversely, when analyzing asymptotic performance, we ablate one factor by removing it entirely while varying the other, so as to evaluate the agent's ability to discover optimal actions absent from the offline prior. The $25\%$ offline data ratio is held constant across all tasks and seeds for these experiments.

We observe on kitchen-partial, kitchen-mixed, and pen-binary that retaining offline data and maintaining the conservative regularizer $\alpha$ both mitigate catastrophic failure, whereas keeping $\alpha$ suppresses asymptotic performance (Figure 3). When offline data is fully removed, final returns are typically lower than when using a fixed mini-batch composition of $25\%$ offline samples at every update. In contrast, annealing the offline fraction to zero, as in our method (Section 7.1), yields faster convergence and final performance comparable to retaining offline data. As highlighted by WSRL (Zhou et al., 2025), persisting offline data during online fine-tuning can depress asymptotic performance, especially when the online and offline distributions are mismatched or the offline data are of lower quality, and also incurs storage/throughput overhead. Hence, if one can avoid using offline data online without sacrificing performance, avoiding offline data online is preferable.

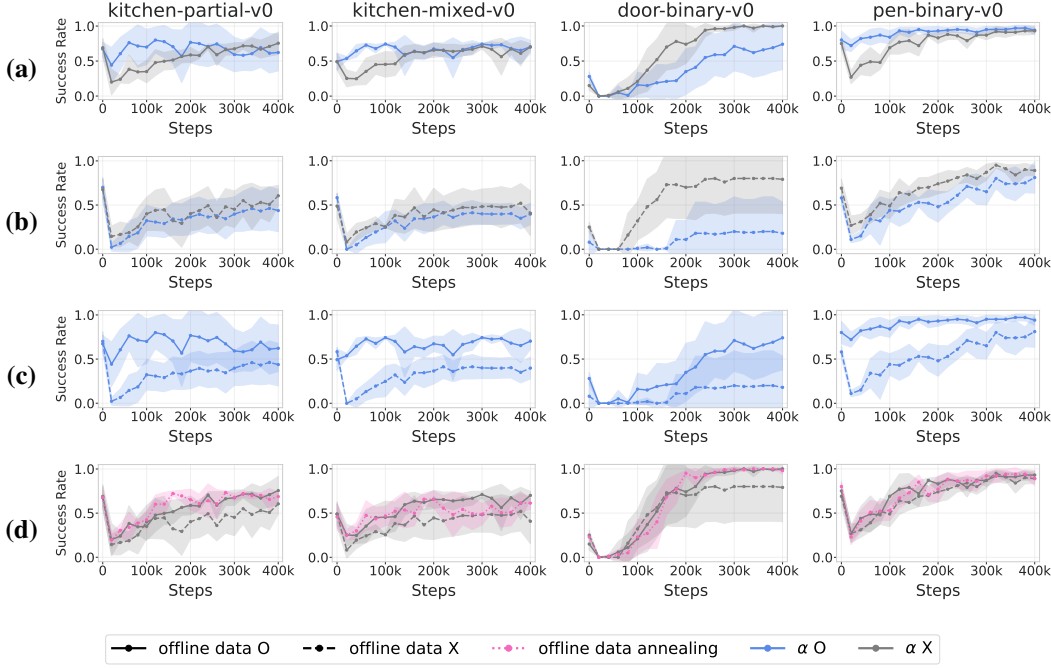

Figure 3: **Controlled analysis of offline data and conservative regularization.** Success rates of CQL during online fine-tuning across four tasks (kitchen-partial, kitchen-mixed, door-binary, pen-binary). (a) Keeping the conservative regularizer ($\alpha$) reduces early collapses. (b) Keeping $\alpha$ lowers asymptotic performance. (c) Dropping offline data induces severe early drops. (d) Annealing offline data speeds convergence; with a suitable schedule, performance can match retaining offline data.

In the door-binary, the low initial success rate makes it difficult to meaningfully compare the effect of each factor on catastrophic failure. However, the conclusion regarding success rates remains consistent with the other tasks. These findings suggest that the inductive bias inherited from offline RL, as well as the continued reliance on offline data during online training, hinder effective exploration of the optimal policy. Thus, while mechanisms to mitigate catastrophic failure remain necessary, removing these constraints is essential for achieving higher asymptotic performance.

## 6 SPURIOUS Q-OPTIMISM AS A DRIVER OF CATASTROPHIC FAILURE

Why does removing the conservative regularizer and offline data in CQL trigger catastrophic failure during online fine-tuning? We posit a single overarching mechanism: *Spurious Q-Optimism*. Early in fine-tuning, the critic can erroneously reverse the relative value ordering between the offline-pretrained policy's action and the action proposed by the current policy for the same state, which steers learning toward actions that later prove inferior under the converged critic. In this section, we (i) formalize this phenomenon, (ii) show that its incidence tracks catastrophic failure across tasks, and (iii) demonstrate that neither tuning hyperparameter in baselines nor slowing the critic's drift from its offline initialization reliably prevents collapse.

### 6.1 QUANTIFYING SPURIOUS Q-OPTIMISM

To test this mechanism, we quantify spurious Q-optimism at each online step via a preference sign mismatch between the current and converged critics. For a minibatch of states $s$, define $\Delta_t(s) = Q_t(s, a_{\mathrm{curr}}) - Q_t(s, a_{\mathrm{off}})$, $\Delta_{\mathrm{final}}(s) = Q_{\mathrm{final}}(s, a_{\mathrm{curr}}) - Q_{\mathrm{final}}(s, a_{\mathrm{off}})$, where $a_{\mathrm{curr}}$ is sampled from the policy at step $t$ and $a_{\mathrm{off}}$ from the offline-pretrained policy on the same states. A state is flagged as optimistic if the signs disagree, equivalently, when $\Delta_t(s)\,\Delta_{\mathrm{final}}(s) < 0$. The *Spurious Q-Optimism Ratio (SQOR)* is the fraction of batch states satisfying this sign mismatch at step $t$.

To analyze the association between SQOR and catastrophic failure, we consider the two stress settings that induce failures in Figure 3: (i) removing offline data during fine-tuning and (ii) removing

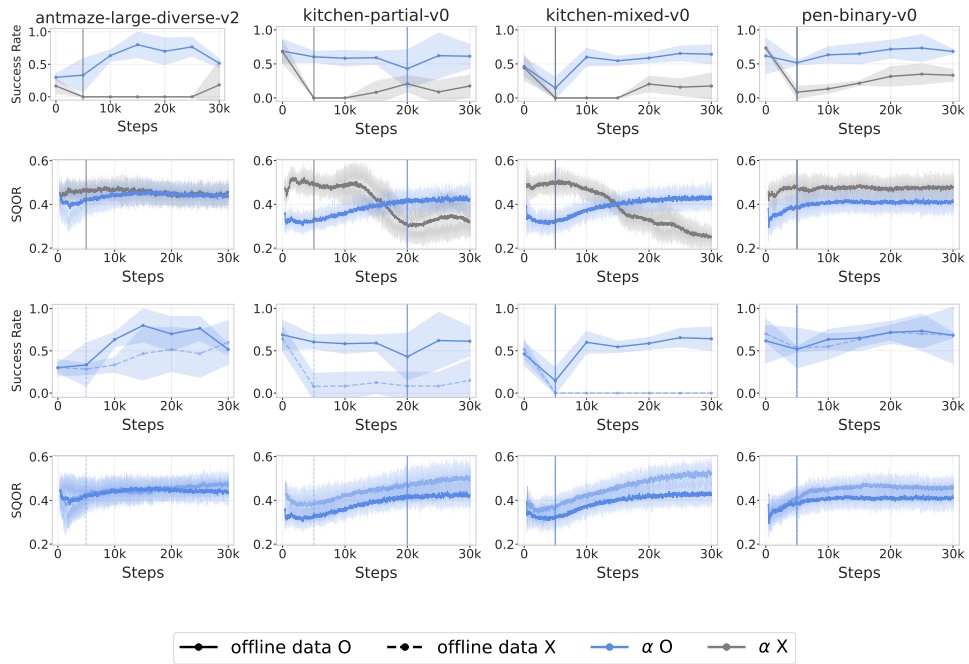

Figure 4: **SQOR tracks catastrophic failure across tasks.** Lower Spurious Q-Optimism Ratio (SQOR) consistently aligns with fewer and milder collapses across all four tasks. Vertical lines mark the onset of catastrophic failure for each method.

the conservative regularizer $\alpha$. Because door-binary exhibits a very low initial success rate and does not manifest a meaningful failure drop (Figure 3), we report results on four tasks, antmaze-large-diverse, kitchen-partial, kitchen-mixed, and pen-binary.

To obtain $Q_{\text{final}}$, we use the checkpoint after $400\text{K}$ online steps for the same random seed, since success rates are stable by that point (Figure 3). Because catastrophic failure emerges within $<$ $30\text{K}$ steps (Figure 3), we report SQOR over the first $30\text{K}$ steps to capture onset dynamics while keeping computation tractable. Due to computational constraints, all experiments analyzing SQOR are conducted with three random seeds.

Empirically, SQOR exhibits a strong correlation with catastrophic failure. As shown in Figure 4, across four tasks and both stress settings, lower SQOR coincides with fewer and milder collapses. In Appendix A, we further examine related diagnostics, the *Spurious Q-Optimism Gap (SQOG)* (aggregate magnitude of preference mismatch), *Online-only SQOR (O-SQOR)* (counts only cases with $\Delta_t(s) > 0$ and $\Delta_{\text{final}}(s) < 0$), and *volatility* (step-to-step Q-value fluctuations). None of these alternatives consistently explain failures across all tasks.

This pattern suggests that the count of misordered state-action comparisons (SQOR) is the primary predictor of collapse. SQOR captures (i) *direct errors*, where the current policy is pulled toward actions that ultimately underperform the offline-pretrained actions (the immediate trigger of collapse), and (ii) an *indirect effect*, where unusually high values assigned to offline-pretrained actions reveal critic instability, although not a direct cause when those actions are selected, such instability can precipitate future direct errors. The fact that O-SQOR (which removes the indirect component) fails to account for failures while SQOR does indicates that this indirect effect materially contributes to catastrophic failure. Formal definitions and full correlation analyses are provided in Appendix A.

## 6.2 CAN CATASTROPHIC FAILURE BE MITIGATED BY HYPERPARAMETER TUNING?

We next investigate whether catastrophic failure can be mitigated solely through tuning hyperparameters in baselines when both $\alpha$ and offline data are removed during online fine-tuning with CQL. One candidate is the UTD ratio, which increases the frequency of critic updates per online interaction and was suggested by Xiao et al. (2025) as a means to alleviate catastrophic failure. However,

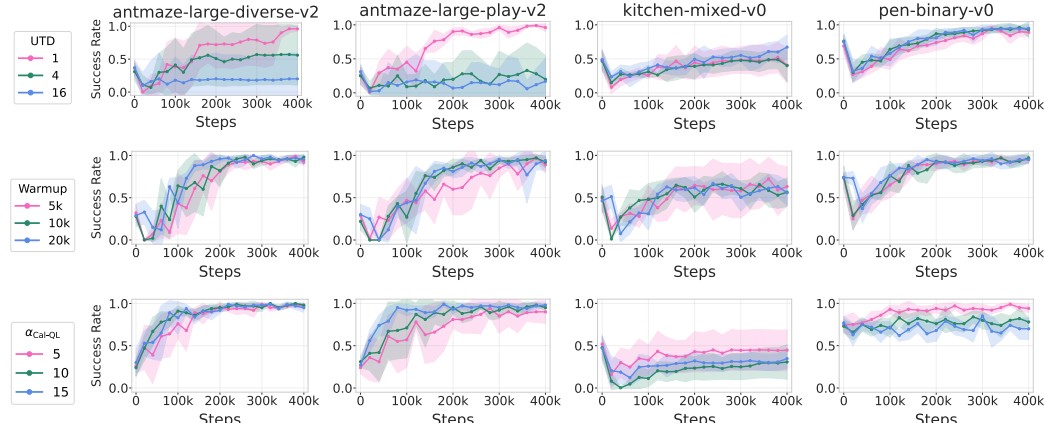

Figure 5: **Hyperparameter tuning does not prevent catastrophic failure.** Effects of varying hyperparameters on online fine-tuning across antmaze-large-diverse, antmaze-large-play, kitchen-mixed, and pen-binary. *Top*: UTD ratio; *Middle*: warmup length; *Bottom*: Cal-QL's conservative-regularizer weight ($\alpha_{\text{Cal-QL}}$). Increasing UTD, extending warmup, or tuning $\alpha_{\text{Cal-QL}}$ fails to avert early performance collapse.

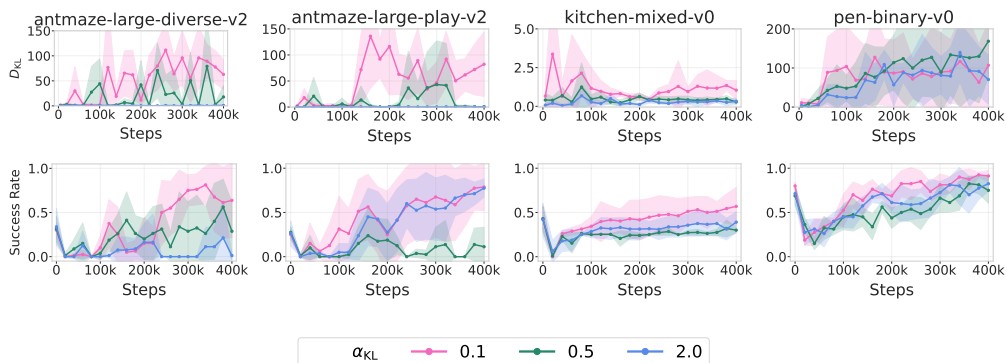

Figure 6: **Regularizing the critic's shift does not prevent collapse.** *Top*: Critic distribution shift during online fine-tuning, measured as $D_{\text{KL}}\big(\text{softmax}(Q^{\text{offline}})|\text{softmax}(Q^{\text{online}})\big)$ under varying KL-penalty strengths $\alpha_{\text{KL}}$ (larger $\alpha_{\text{KL}}$ slows the shift). *Bottom*: Corresponding success rates. Across four tasks (antmaze-large-diverse, antmaze-large-play, kitchen-mixed, pen-binary), even when the shift is substantially reduced, early-stage catastrophic failure persists.

as shown in Figure 5, simply increasing the UTD ratio does not effectively reduce catastrophic failure across tasks. We also evaluated the use of a warmup phase, where the agent does not update its parameters but collects trajectories via online interaction using the offline-pretrained policy, as proposed in Zhou et al. (2025). Increasing the warmup length likewise fails to reduce catastrophic failure across tasks.

In addition, Cal-QL proposed modifying the conservative regularizer in CQL as a potential remedy (Nakamoto et al., 2023). We experimented with retaining the regularizer while varying its weight ($\alpha_{\text{Cal-QL}}$). However, increasing the weight did not produce meaningful reductions in catastrophic failure across tasks. This indicates that even the modified conservative regularizer in Cal-QL does not provide a significant mitigation effect.

Beyond hyperparameter tuning, we also examine whether moderating the critic's distributional shift away from the offline-pretrained critic can alleviate catastrophic failure. To control the shift speed, we augment the TD loss with a KL penalty between the action distributions induced by the offline and online critics, optimizing

$$\min_{\theta} \mathcal{L}_{\text{TD}}(\theta) + \alpha_{\text{KL}} \mathbb{E}_{s,a\sim\mathcal{B}} \big[ D_{\text{KL}} \big( \text{softmax}\big(Q_\theta^{\text{offline}}(s,\cdot)\big)|\text{softmax}\big(Q_\theta^{\text{online}}(s,\cdot)\big)\big)\big],$$

where $\mathcal{B}$ denotes the replay buffer and $\alpha_{\mathrm{KL}} > 0$ controls the strength of the KL penalty, with larger values more strongly suppressing the distributional shift, as empirically illustrated in Figure 6. This objective is motivated by the hypothesis of Zhou et al. (2025) that slowing the critic's distributional shift may reduce catastrophic failure. However, Figure 6 shows that in practice, more aggressive regularization of the critic's shift does not reduce catastrophic failure.

Further analyses of hyperparameter adjustments, including batch size, network dimensions, and learning rates, are presented in Appendix C. None of these adjustments produced substantial improvements across tasks, reinforcing our claim that Spurious Q-Optimism is the fundamental driver of catastrophic failure.

# 7 ANNEALING BRIDGES OFFLINE AND ONLINE RL

Building on the analyses in Sections 5 and 6, we propose **SOAR** (*Smooth Offline-to-Online Annealing for RL*), a simple yet effective method that mitigates catastrophic failure while attaining superior asymptotic performance. The core idea is to gradually remove conservative regularizer and offline data that stabilize early fine-tuning but hinder long-term performance. In addition, through extensive ablations, we disentangle the respective roles of offline data and $\alpha$ annealing, and show how practitioners can tailor SOAR's components as practical design choices depending on whether robustness or final performance is prioritized.

## 7.1 SOAR (SMOOTH OFFLINE-TO-ONLINE ANNEALING FOR RL)

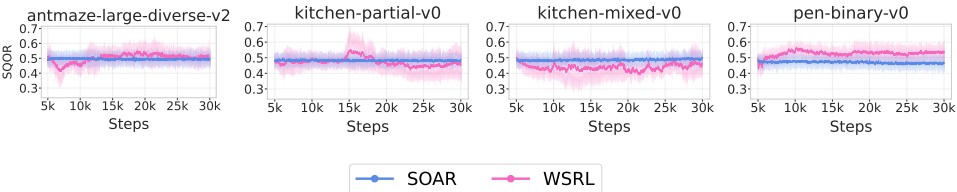

Figure 7: **WSRL vs. SOAR on SQOR.** Spurious Q-Optimism Ratio (SQOR) over the early online phase across four tasks. SQOR trajectories for WSRL and SOAR are statistically indistinguishable on antmaze-large-diverse, kitchen-partial, and kitchen-mixed, whereas WSRL exhibits higher SQOR on pen-binary, aligning with its larger catastrophic failure in Figure 2.

SOAR uses standard CQL for offline pre-training on $\mathcal{D}_{\mathrm{off}}$ and modifies only the online fine-tuning phase. The key idea is to gradually remove (i) the offline data used in replay and (ii) the conservative regularizer, to stabilize at the beginning of fine-tuning while enabling unconstrained exploration later. Concretely, at online step $t$ we form a minibatch $\mathcal{B}_t$ by mixing samples from the offline and online buffers, $\mathcal{B}_t \sim \lambda_t \mathcal{D}_{\mathrm{off}} + (1 - \lambda_t) \mathcal{D}_{\mathrm{on}}$, where $\lambda_t \in [0, 1]$ is a replay composition schedule that monotonically decreases to $0$. The critic is updated with the usual CQL objective, but with a time-varying conservative weight $\alpha_t$:

$$\mathcal{L}_t^{\mathrm{SOAR}}(\theta) = \mathcal{L}_{\mathrm{TD}}(\theta; \mathcal{B}_t) + \alpha_t \Big( \mathbb{E}_{s \sim \mathcal{B}_t, a \sim \pi_\theta(\cdot|s)}[Q_\theta(s, a)] - \mathbb{E}_{(s,a) \sim \mathcal{B}_t}[Q_\theta(s, a)] \Big), \qquad (2)$$

and the actor is optimized as in CQL with no architectural changes. Thus, SOAR is algorithmically minimal: it introduces no new losses beyond CQL, only two annealing schedules.

We employ a linear schedule for the offline data-ratio, $\lambda_t = \max\{0, \ \lambda_0(1 - t/T_\lambda)\}$, so that the fraction of offline samples decays monotonically to zero, and an exponential schedule for the conservative weight with separate controls for decay rate and annealing interval: $\alpha_t = \max\{0, \ \alpha_0 \exp\big(- r \cdot (t/T_\alpha)\big)\}$. Here, $\lambda_0$ denotes the initial offline data fraction, $T_\lambda$ the annealing horizon for replay composition, $\alpha_0$ the conservative weight used during offline pre-training, $r > 0$ the decay rate, and $T_\alpha > 0$ the annealing interval. Task-specific settings for $(\lambda_0, T_\lambda, \alpha_0, r, T_\alpha)$ are provided in Appendix G. Early in fine-tuning, larger $\lambda_t$ and $\alpha_t$ damp catastrophic failure by anchoring to offline support and pessimism; as $t$ increases, both terms vanish, allowing the agent to fully exploit online interaction. We provide the pseudocode for SOAR in Algorithms 1.

Despite its simplicity, SOAR consistently reduces catastrophic failure and improves asymptotic performance across tasks (Figure 2). Examining the diagnostics, SQOR levels for SOAR and WSRL

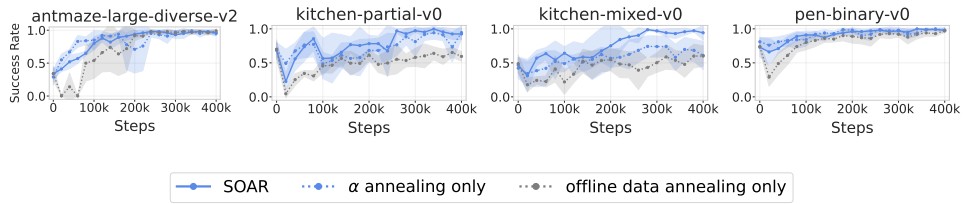

Figure 8: **Single-component annealing ablations.** Comparison of (i) conservative-weight ($\alpha$) annealing only and (ii) offline data-ratio annealing. Offline data-ratio annealing reduces early performance collapse but can limit final success rate, while combining both (SOAR) achieves the best overall performance across all four tasks.

are statistically indistinguishable on antmaze-large-diverse, kitchen-partial, and kitchen-mixed (Figure 7). However, on pen-binary, WSRL exhibits a significantly higher SQOR than SOAR, mirroring the relative magnitude of catastrophic failure observed in Figure 2.

Crucially, the dual annealing schedules enable a complete phase-out of both the offline replay (data ratio $\lambda_t \to 0$) and the conservative penalty (weight $\alpha_t \to 0$) during fine-tuning, yielding a smooth transition to fully online, non-conservative training without the large performance collapses observed in prior methods. For complete numerical results across all tasks, see Appendix K.

### 7.2 ABLATION: WHAT DOES EACH ANNEALING COMPONENT CONTRIBUTE?

To quantify the contribution of each component in SOAR, we ablate the dual annealing design into two single-component variants and compare them against the full method. (i) Offline data-ratio annealing only: $\lambda_t$ decays linearly to 0 while the conservative penalty is disabled throughout online fine-tuning ($\alpha_t = 0$). (ii) Conservative-weight ($\alpha$) annealing only: $\alpha_t$ decays exponentially to 0 while no offline data are mixed into replay ($\lambda_t = 0$). For the scheduling in each variant, we use the same hyperparameters as in SOAR. We also report ablation on the antmaze-ultra-diverse task which is a harder version of antmaze-large-diverse, in Appendix D.

Results in Figures 8 and 13 indicate a task-dependent trade-off. The preferred annealing depends on the severity of exploration-induced collapse and on whether risk mitigation or asymptotic return is prioritized. When exploration risk is severe, retaining conservatism while annealing only the data ratio tends to yield the most stable learning. Otherwise, as a robust default across tasks, using both annealings (SOAR) provides the most reliable risk-return balance.

## 8 CONCLUSION

We introduced **SOAR**, a simple offline-to-online fine-tuning procedure that jointly anneals the offline replay ratio and the conservative regularizer. This dual schedule balances two competing objectives, reducing early catastrophic failure and achieving strong asymptotic performance, by retaining offline data and pessimism at the start of fine-tuning and then smoothly phasing both out to enable unconstrained online improvement.

Our analysis identifies *spurious Q-optimism* as a primary driver of collapse: early critics can misorder the current-policy and offline-pretrained actions for the same state, and the resulting *Spurious Q-Optimism Ratio* (SQOR) closely tracks catastrophic failures across tasks. In contrast, a range of alternatives, including hyperparameter variations, warmup, higher UTD ratios, and explicit penalties that slow the critic's drift from its offline initialization, do not reliably prevent collapse. Ablations further show that while single-component schedules (only data-ratio or only conservative-weight annealing) offer useful knobs for safety or speed, their combination (SOAR) delivers the most favorable stability-performance trade-off overall.

Beyond a practical recipe, SOAR offers a lens on offline-to-online RL: the transition is best viewed as balancing stability against exploration. This suggests several directions for future work, including adaptive or performance-aware scheduling, tighter theory linking spurious Q-Optimism to failure probabilities, and extensions to vision-based and real-robot settings. We hope these findings encourage more principled designs for bridging offline pre-training and online improvement.

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

# A    ADDITIONAL ANALYSIS ON CATASTROPHIC FAILURE

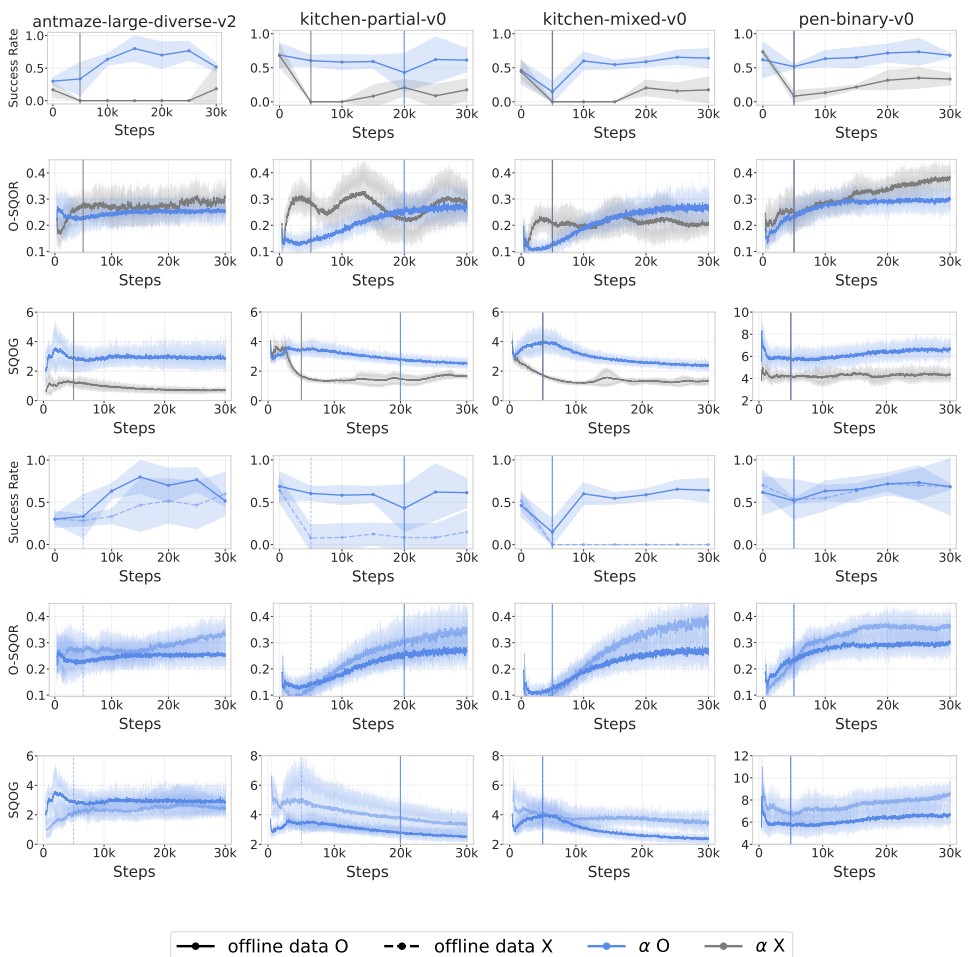

Figure 9: **SQOG and O-SQOR do not predict collapse.** Across four tasks and two stress settings, SQOG (magnitude of mismatch) and O-SQOR (one-sided optimism) show inconsistent alignment with catastrophic failures.

In Section 6, we showed that the *Spurious Q-Optimism Ratio* (SQOR) exhibits a strong correlation with catastrophic failure. Could other factors be responsible? In this section, we systematically investigate alternative explanations and find none that consistently account for the observed failures. Mirroring the SQOR analysis, we examine correlations between catastrophic failure and candidate metrics under two stress settings, removing offline data and removing the conservative regularizer $\alpha$, across four tasks, antmaze-ultra-diverse, kitchen-partial, kitchen-mixed, and pen-binary.

**Spurious Q-Optimism Gap (SQOG).** Let $\Delta_t(s) = Q_t(s, a_{\mathrm{curr}}) - Q_t(s, a_{\mathrm{off}})$ and $\Delta_{\mathrm{final}}(s) = Q_{\mathrm{final}}(s, a_{\mathrm{curr}}) - Q_{\mathrm{final}}(s, a_{\mathrm{off}})$. With $\mathcal{M}_t = \{\, s : \mathrm{sgn}(\Delta_t(s)) \neq \mathrm{sgn}(\Delta_{\mathrm{final}}(s)) \,\}$, we define

$$\mathrm{SQOG}(t) = \frac{1}{|\mathcal{M}_t|} \sum_{s \in \mathcal{M}_t} \big| \Delta_t(s) - \Delta_{\mathrm{final}}(s) \big|.$$

SQOG measures the *magnitude* of preference mismatch on misordered states. As shown in Figure 9, SQOG does not exhibit a consistent trend with catastrophic failure across tasks, suggesting that the *number* of misordered states (captured by SQOR) is more predictive of collapse than the size of a few large errors.

**Online Spurious Q-Optimism Ratio (O-SQOR).** Focusing on the one-sided, harmful optimism cases, we define

$$\text{O-SQOR}(t) = \frac{1}{|\mathcal{B}_t|} \sum_{s \in \mathcal{B}_t} \mathbf{1}\Big[\Delta_t(s) > 0 \ \wedge \ \Delta_{\text{final}}(s) < 0\Big],$$

i.e., the current critic prefers the current-policy action, while the converged critic prefers the offline action. O-SQOR does not consistently align with catastrophic failure across tasks than SQOR (Figure 9), indicating that both mismatch directions contribute to overall collapse risk.

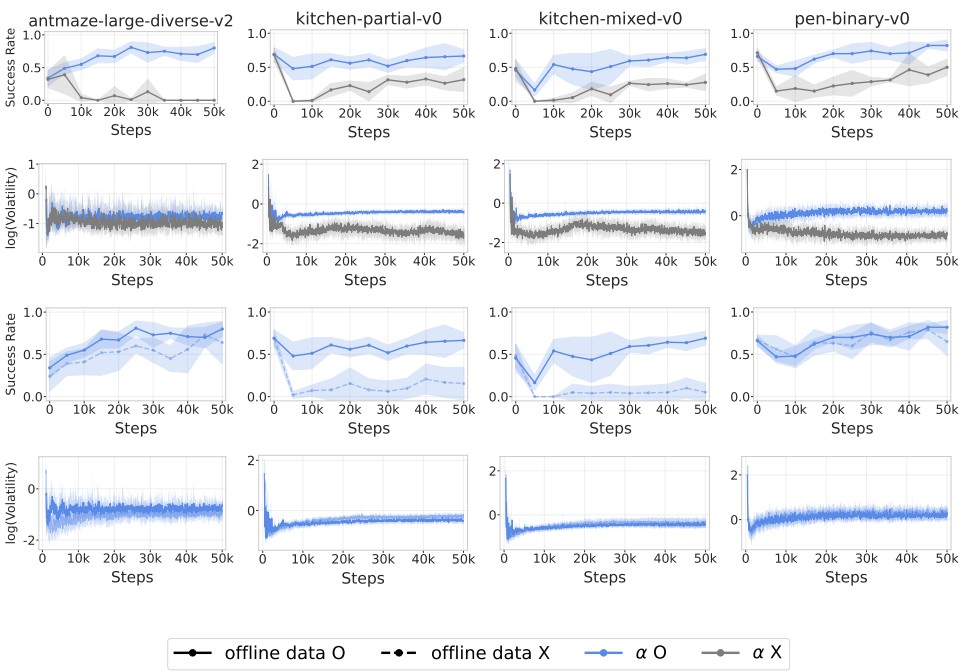

Figure 10: **Volatility does not predict collapse.** Across tasks and stress settings, step-to-step Q-update volatility shows weak and inconsistent correspondence with catastrophic failure.

**Volatility.** We define volatility as the square root of the bias-corrected exponential moving average of squared one-step Q-updates (Kingma & Ba, 2014):

$$\text{Volatility}_t = \sqrt{\frac{m_t}{1 - \beta^t}}, \quad m_t = \beta\, m_{t-1} + (1 - \beta)\, \mathbb{E}_{(s,a) \sim \mathcal{B}_t}\big[\big(\Delta Q_t(s,a)\big)^2\big],$$

where $\Delta Q_t(s,a) = Q_{t+1}(s,a) - Q_t(s,a)$ over minibatch $\mathcal{B}_t$, with $\beta = 0.9$ in all experiments. Intuitively, volatility measures the magnitude of per-pair $(s,a)$ Q-value adjustments; larger values indicate more abrupt updates. As shown in the fourth row of Figure 10, volatility does not consistently track catastrophic failure across tasks, suggesting that SQOR is more predictive of collapse than aggregate fluctuations of $Q$ over all $(s,a)$ pairs.

# B ADDITIONAL ABLATION STUDIES ON SOAR

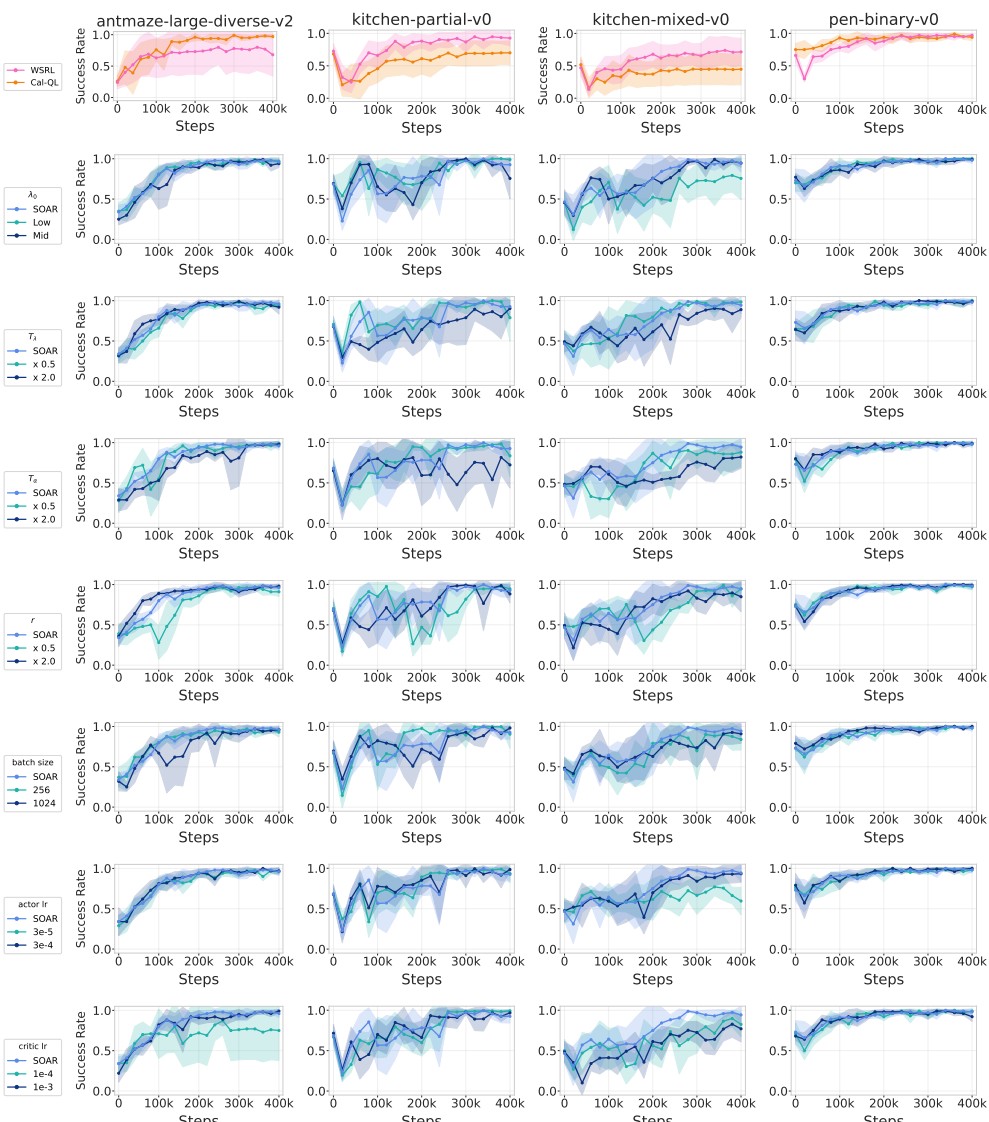

Figure 11: **SOAR ablations.** Sensitivity to (a) initial offline data-ratio, (b) offline data-ratio annealing interval, (c) conservative-weight ($\alpha$) annealing interval, and (d) $\alpha$ annealing temperature. (e) batch size, (f) actor learning rate, (g) critic learning rate. Across tasks, SOAR remains robust, exhibiting low catastrophic failure and strong asymptotic success over a wide range of settings.

We examine how each hyperparameter influences SOAR's behavior, with the goal of understanding the role of each component and the method's sensitivity. Unless stated otherwise, exact values and ranges are listed in Appendix G. Figure 12 summarizes results on four benchmarks: antmaze-large-diverse, kitchen-partial, kitchen-mixed, and pen-binary.

**Initial offline-data ratio.** We vary the initial fraction of offline samples mixed into replay before annealing, testing starting ratios of 0.25 (low) and 0.5 (mid). SOAR is generally robust to this choice: antmaze-large-diverse and pen-binary show negligible changes in both early stability and final return across settings. On kitchen-partial, starting from a smaller ratio reduces the initial dip while preserving asymptotic performance, indicating headroom for further improvements via more aggressive early down-weighting. On kitchen-mixed, a small initial ratio hurts both early stability and final success, although it still outperforms the strongest baseline.

**Offline data-ratio annealing interval.** Shorter schedules hasten the transition to fully online training and tend to speed convergence on kitchen-partial and kitchen-mixed, though variance increases on the latter. Lengthening the interval delays the transition and can cap the final return on kitchen-mixed, while yielding similar asymptotic performance on the other tasks. A moderate schedule offers a favorable stability-performance trade-off.

**Conservative-weight ($\alpha$) annealing interval.** Doubling $T_\alpha$ induces greater instability during training on kitchen-partial and kitchen-mixed and yields lower asymptotic performance. Conversely, shortening $T_\alpha$ preserves final performance but increases variance, underscoring the importance of choosing an appropriate interval.

**$\alpha$ annealing temperature.** Faster exponential decay improves convergence on antmaze-large-diverse but exacerbates early collapse on kitchen-mixed. Slower decay mainly postpones the timing of collapse on kitchen-partial and kitchen-mixed rather than reducing its magnitude, and can worsen collapse on antmaze-large-diverse.

**Batch size.** Increasing batch size in antmaze raises variance during learning but, overall, both minima and final plateaus change little across tested values, indicating SOAR's gains are not contingent on batch size tuning.

**Actor learning rate.** Varying the actor step size within a standard range produces only minor differences on most tasks. Very small rates slow progress; very large rates can introduce transient oscillations. The default strikes a good speed-stability balance, and nearby values behave similarly.

**Critic learning rate.** Patterns mirror the actor: very small rates slow learning (e.g., antmaze-large-diverse and kitchen-partial), while very large rates can cause oscillations (kitchen-partial). The default again provides a reasonable compromise.

**Takeaway.** While the magnitude of catastrophic failure and the ultimate success rate can shift with hyperparameters, SOAR variants consistently matches or outperform baselines across settings. Batch size and learning rates have comparatively modest effects, whereas annealing schedules materially influence both early stability and asymptotic performance. Together with the single-component analyses in Section 7.2, these results suggest that practitioners can tailor design choices on the method to their safety and performance requirements while retaining SOAR's core benefits.

## C OTHER ATTEMPTS TO MITIGATE CATASTROPHIC FAILURE

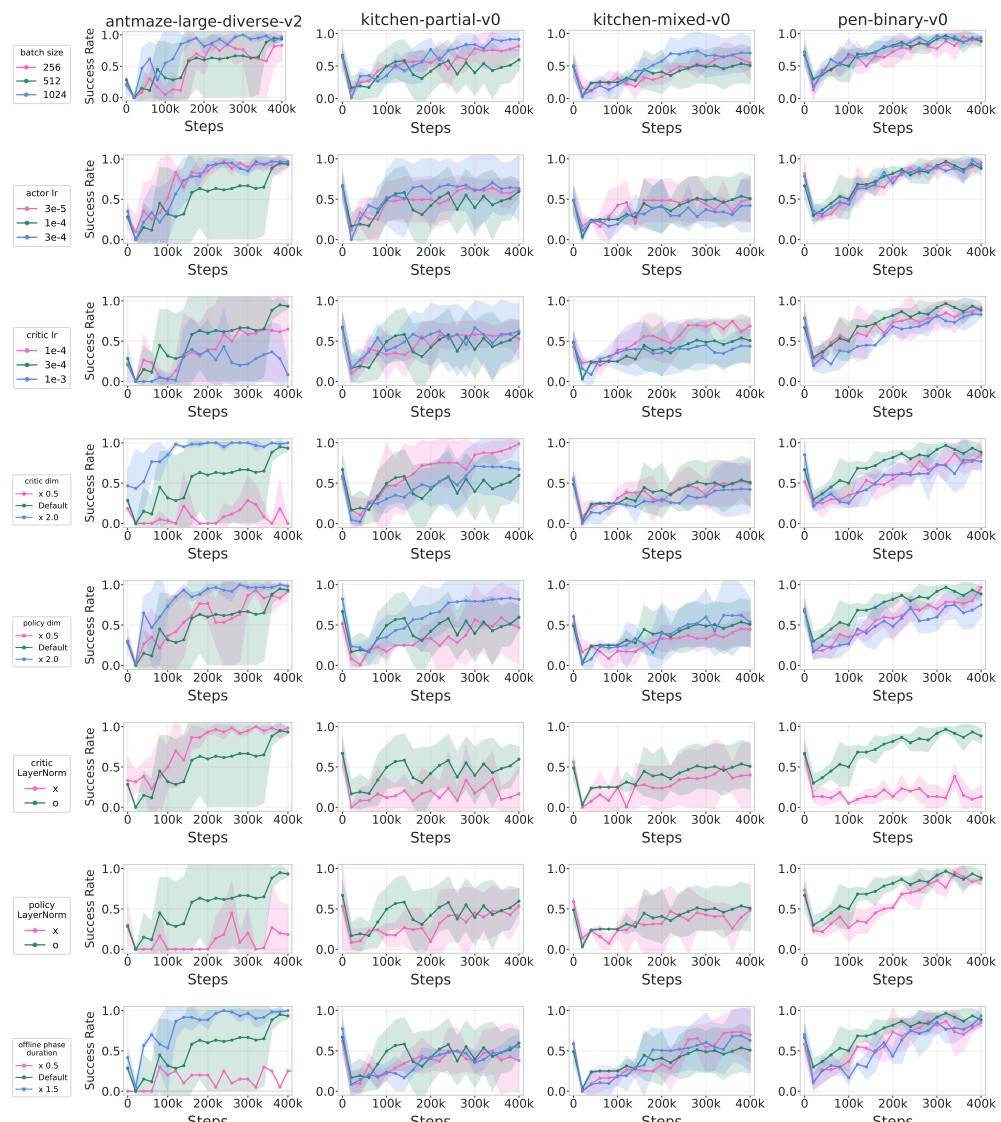

Figure 12: **Hyperparameter and architectural tweaks do not consistently mitigate catastrophic failure.** From top to bottom, the plots correspond to: batch size; actor learning rate; critic learning rate; critic hidden dimension; actor hidden dimension; LayerNorm in the critic; LayerNorm in the actor; and the duration of the offline pre-training phase. Across all settings and tasks, none of these factors consistently mitigates catastrophic failure.

In Section 6.2, we examined whether tuning the UTD ratio, varying warmup length, adjusting the Cal-QL conservative weight $\alpha_{\text{Cal-QL}}$, or regularizing the critic shift via $D_{\text{KL}}\big(\text{softmax}(Q^{\text{offline}})|\text{softmax}(Q^{\text{online}})\big)$ could mitigate catastrophic failure. None of these interventions proved effective. Here, we extend this investigation to additional settings: batch size, actor learning rate, critic learning rate, critic layer hidden dimension size, actor layer hidden dimension size, LayerNorm in the critic, LayerNorm in the actor, and the duration of the offline pre-training phase. As shown in Figure 12, across four tasks none of these choices consistently reduces catastrophic failure, reinforcing our conclusion that such collapses are not resolved by routine hyperparameter or architectural tweaks.

## D ANTMAZE-ULTRA-DIVERSE: A HIGH-RISK STRESS TEST FOR OFFLINE-TO-ONLINE RL

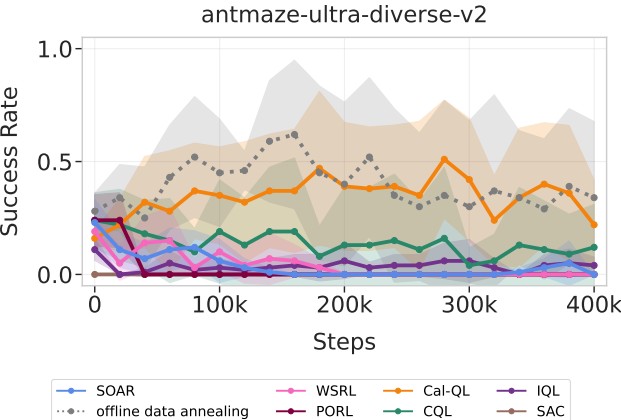

Figure 13: **AntMaze-Ultra-Diverse results.** This highlights the value of retaining a conservative regularizer when exploration carries high risk.

AntMaze-Ultra-Diverse presents an extreme exploration regime with highly diverse start-goal pairs and long horizons, where online rollouts readily drift off the offline support, and sparse rewards make recovery difficult. Figure 13 summarizes the results. Purely online SAC stays near zero, and standard offline baselines (CQL, IQL) remain low and flat. Offline-to-online methods that remove offline inductive bias early (WSRL, PORL, and our dual annealing SOAR) also struggle to make headway in this setting. In contrast, the variant that anneals offline data while fixing conservatism, decaying the replay ratio $\lambda_t$ but keeping the conservative weight $\alpha$ unchanged, delivers the most stable learning and competitive final returns here.

This suggests a practical design rule: when exploration is liable to precipitate extreme failure, it can be preferable to retain conservative regularizer during fine-tuning, e.g., anneal $\lambda_t$ while holding (or very slowly annealing) $\alpha$, trading some exploration freedom for robustness. On easier tasks, our main results show that dual annealing (SOAR) typically offers the best overall trade-off.

## E EXTENDED RELATED WORKS

**Offline RL.** Online RL requires actual interaction with the environment, which can be expensive or dangerous in domains like robotics, healthcare, and recommender systems (Singh et al., 2022; Liu et al., 2020; Chen et al., 2024). Offline RL emerged to mitigate this issue by using a static dataset previously collected by a behavior policy to enable sample-efficient policy learning (Prudencio et al., 2023). While effective, a policy trained with offline static data faces significant problems when it encounters unseen circumstances (Prudencio et al., 2023). Various approaches have been proposed to address this limitation, which can be categorized into regularization (Fujimoto & Gu, 2021; Park et al., 2025; Tarasov et al., 2023; Kumar et al., 2020), uncertainty estimation (An et al., 2021; Nikulin et al., 2023), model-based methods (Kidambi et al., 2020; Yu et al., 2020; 2021; Chitnis et al., 2024; Park & Lee, 2024), one-step methods (Brandfonbrener et al., 2021; Eysenbach et al., 2022; Park et al., 2025), weighted regression (Peng et al., 2019; Wang et al., 2020) and in-sample maximization (Garg et al., 2023; Kostrikov et al., 2022; Xu et al., 2023). Among these, we build upon CQL (Kumar et al., 2020) as the foundation for our method, which directly addresses the overestimation issue by adding a penalty to the Q-function. We adopted IQL (Kostrikov et al., 2022) as one of our baselines, which improves upon the behavior policy in the dataset by leveraging the value function's generalization to enhance the quality of policy, without evaluating out-of-distribution actions.

# F  DETAILS ON BASELINE ALGORITHMS

**Soft Actor-Critic (SAC)** (Haarnoja et al., 2018). SAC extends standard actor-critic methods by incorporating entropy maximization into the RL objective. Instead of maximizing only the expected return, SAC also maximizes the entropy of the policy to encourage exploration and prevent premature convergence to deterministic policies. The resulting objective is to maximize the expected sum of rewards and entropies:

$$J(\pi) = \mathbb{E}_\pi \left[ \sum_{t=0}^\infty \gamma^t \left( r(s_t, a_t) + \alpha \mathcal{H}(\pi(\cdot|s_t)) \right) \right],$$

where $\mathcal{H}(\pi(\cdot|s_t)) = -\mathbb{E}_{a_t \sim \pi}[\log \pi(a_t|s_t)]$ denotes the entropy of the policy at state $s_t$, and $\alpha > 0$ is the temperature parameter that balances reward maximization and entropy.

**Conservative Q-Learning (CQL)** (Kumar et al., 2020). CQL learns a Q-function by explicitly regularizing Q-values to mitigate overestimation issues common in offline RL (Prudencio et al., 2023):

$$\mathcal{L}(\theta) = \alpha \underbrace{\left( \mathbb{E}_{s \sim \mathcal{D}, a \sim \pi}[Q_\theta(s, a)] - \mathbb{E}_{s, a \sim \mathcal{D}}[Q_\theta(s, a)] \right)}_{\text{Conservative regularizer}} + \mathcal{L}_{\text{TD}}(\theta)$$

Here, $\pi$ represents the current policy, $\mathcal{D}$ is the offline dataset, and $\alpha$ controls the intensity of the conservative regularization. The term $\mathcal{L}_{TD}$ is the temporal difference loss used in Q-learning methods, while the conservative regularization penalizes Q-values for state-action pairs not present in the offline dataset $\mathcal{D}$.

**Implicit Q-learning (IQL)** (Kostrikov et al., 2022). The IQL algorithm learns a state-value function $V_{\theta_V} : \mathcal{S} \to \mathbb{R}$ and an action-value function $Q_{\theta_Q} : \mathcal{S} \times \mathcal{A} \to \mathbb{R}$ by minimizing:

$$\mathcal{L}_V(\theta_V) = \mathbb{E}_{(s,a) \sim \mathcal{D}} \left[ \ell_\kappa^2 \left( V_{\theta_V}(s) - Q_{\bar{\theta}_Q}(s, a) \right) \right],$$

$$\mathcal{L}_Q(\theta_Q) = \mathbb{E}_{(s,a,r,s') \sim \mathcal{D}} \left[ \left( Q_{\theta_Q}(s, a) - r - \gamma V_{\theta_V}(s') \right)^2 \right],$$

where the expectile loss is defined as $\ell_\kappa^2(x) = |\kappa - \mathbb{1}[x < 0]|x^2$ (Newey & Powell, 1987), and $\bar{\theta}_Q$ represents the target network parameters. Unlike standard MSE loss, expectile loss asymmetrically weights positive and negative errors, with $\kappa > 0.5$ emphasizing higher Q-values. By doing so, IQL evaluates only actions from the dataset, which enables policy improvement without querying out-of-distribution actions, thereby inducing implicit conservatism.

**Calibrated Q-Learning (Cal-QL)** (Nakamoto et al., 2023). Cal-QL adjusts CQL by calibrating the learned Q-function relative to a reference policy to avoid initial degradation caused by overly pessimistic Q-values during fine-tuning. The conservative regularization term in CQL is altered by substituting $\mathbb{E}_{s \sim \mathcal{D}, a \sim \pi}[Q_\theta(s, a)]$ with $\mathbb{E}_{s \sim \mathcal{D}, a \sim \pi}[\max(Q_\theta(s, a), V^\mu(s))]$, where $V^\mu(s)$ represents the value function of the reference policy $\mu$, which can be estimated via Monte-Carlo return. This modification prevents the problem of overly small Q-values in CQL by giving a lower bound with reference policy.

**Policy-Only Reinforcement Learning Fine-Tuning (PORL)** (Xiao et al., 2025). PORL focuses on the challenge of fine-tuning online RL using only a pretrained policy, without relying on pretrained Q-functions or offline datasets. This approach is especially beneficial when pretrained Q-functions are either unreliable due to pessimism or unavailable, such as in imitation learning scenarios. PORL begins by training the randomly initialized Q-function at the beginning of the online fine-tuning phase. Training data is collected based on the online interaction of the pretrained policy using an epsilon-greedy exploration strategy. During this initial sampling phase, the Q-function is trained via temporal difference learning with a high UTD ratio. After this pre-sampling period, PORL transitions to standard SAC fine-tuning, updating both the policy and the Q-function.

**Warm-start RL (WSRL)** (Zhou et al., 2025). WSRL addresses the distribution shift issue from offline to online data in fine-tuning without offline data by introducing a warmup phase. This phase utilizes a frozen pretrained policy to collect online rollouts. The data collected during warmup bridges the distribution mismatch and helps recalibrate the offline Q-function to the online distribution, allowing the method to adapt in the online environment quickly. Once the warmup phase is complete, WSRL proceeds with conventional online RL using SAC, employing a high UTD ratio.

# G  IMPLEMENTATION DETAILS

## G.1  DETAILS ON ENVIRONMENTS

**AntMaze**. The AntMaze environment, part of the D4RL benchmark suite (Fu et al., 2020), tasks an 8-DoF ant quadruped robot with navigating through a large and complex maze to reach a designated goal position. The agent receives a binary reward of +1 only upon successfully reaching the goal. The observation space is 29-dimensional, including the robot's position, orientation, and velocity, while the action space is a continuous 8-dimensional vector, normalized to the range $[-1, 1]$. We use two variants: antmaze-large-diverse-v2, which contains trajectories collected by commanding the agent to random goals from random start positions, and antmaze-large-play-v2, which contains trajectories directed to a specific location. Both environments share the same maze structure and maximum episode length of 1000 steps.

**FrankaKitchen**. The FrankaKitchen environment contained in D4RL (Fu et al., 2020) requires controlling a 9-DoF Franka Panda robotic arm to manipulate various kitchen appliances and configure the environment into a predefined target state. Each task consists of four subtasks, and the agent receives a reward between 0 and 4 based on the number of successfully completed subtasks. The observation space is 60-dimensional, encompassing joint positions and object states, and the action space is a 9-dimensional continuous vector normalized to $[-1, 1]$. We use two benchmark environments from D4RL: kitchen-partial-v0 and kitchen-mixed-v0. The former includes a mix of complete and incomplete demonstrations, where task elements involve operating the microwave, kettle, light switch, and slide cabinet. The latter contains only incomplete demonstrations. Its task components include the microwave, kettle, bottom burner, and light switch. Both environments use a maximum episode length of 280 steps.

**Adroit**. The Adroit suite (Rajeswaran et al., 2017) evaluates dexterous manipulation using a high-DoF robotic hand. Specifically, we use two tasks from the D4RL benchmark: pen-binary-v0 and door-binary-v0. In pen-binary-v0, a 24-DoF shadow hand must reorient a pen to match a target pose, while in door-binary-v0, a 28-DoF hand must grasp and rotate a door handle to open it. Both environments have sparse binary rewards: a reward of +1 is given only upon successful task completion. The observation space is 45-dimensional for the pen task and 39-dimensional for the door task, consisting of hand joint angles and object poses. The action space is continuous, 24-dimensional for the pen task and 28-dimensional for the door task, each normalized to $[-1, 1]$. The maximum episode lengths are 100 and 200 steps respectively.

## G.2  NETWORK ARCHITECTURES

The network architecture is adopted directly from WSRL (Zhou et al., 2025), where the agent follows an actor-critic framework. The actor network takes the observation and outputs the mean and the log standard deviation of a Gaussian distribution to sample actions. The critic network takes the concatenated observation and action vectors and feeds them into an ensemble of 10 Q-functions. Each Q-function outputs a scalar Q-value. For stability, 2 Q-functions are randomly subsampled during target computation, and minimum value is used. Both the actor and the critic are implemented as multilayer perceptrons (MLPs) with Rectified Linear Unit (ReLU) activation. A learnable temperature parameter is used to control the entropy regularization. The exact hidden dimensions for each domain are provided in Table 1.

## G.3  HYPERPARAMETERS

Table 1 summarizes the hyperparameters used in our experiments. Unless noted otherwise, optimization and architecture settings follow WSRL (Zhou et al., 2025), with the following uniform modifications across all baselines: (i) the batch size is increased from 256 to 512; (ii) we employ an ensemble of ten Q-functions and compute targets as the minimum over two critics randomly subsampled each update, following REDQ (Chen et al., 2021); and (iii) layer normalization (Ba et al., 2016) is applied to both the policy and critic networks to mitigate uncontrolled extrapolation effects (Ball et al., 2023). For PORL, we adopt the authors' settings described in Xiao et al. (2025).

| | Antmaze | | | Adroit | | Kitchen | |
|---|---|---|---|---|---|---|---|
| | large diverse | large play | ultra diverse | door | pen | partial | mixed |
| **SOAR** | | | | | | | |
| $\lambda_0$ | 0.75 | 0.25 | 0.25 | 0.25 | 0.75 | 0.9 | 0.75 |
| $T_\lambda$ | 40,000 | 40,000 | 40,000 | 20,000 | 40,000 | 80,000 | 20,000 |
| $T_\alpha$ | 80,000 | 80,000 | 40,000 | 40,000 | 80,000 | 160,000 | 160,000 |
| $r$ | 5.0 | 5.0 | 5.0 | 5.0 | | 5.0 | |
| **Optimization** | | | | | | | |
| Actor Learning Rate | | $1 \times 10^{-4}$ | | | $1 \times 10^{-4}$ | $1 \times 10^{-4}$ | |
| Critic Learning Rate | | $3 \times 10^{-4}$ | | | $3 \times 10^{-4}$ | $3 \times 10^{-4}$ | |
| Batch Size | | 512 | | | 512 | 512 | |
| UTD | | 1 | | | 1 | 1 | |
| Offline Steps | | 1,000,000 | | | 40,000 | 250,000 | |
| $\alpha (= \alpha_0)$ | | 5.0 | | | 1.0 | 5.0 | |
| warmup step | | 5,000 | | | 5,000 | 5,000 | |
| **Architecture** | | | | | | | |
| Critic Network | | [256, 256, 256, 256] | | | [512, 512, 512] | [512, 512, 512] | |
| Actor Network | | [256, 256] | | | [512, 512] | [512, 512, 512] | |
| Activations | | ReLU | | | ReLU | ReLU | |
| Q-ensemble | | 10 | | | 10 | 10 | |

Table 1: **Hyperparameters.**

We perform hyperparameter tuning with a particular focus on three key components, $\lambda_0$, $T_\lambda$, and $T_\alpha$. Specifically, we explore $\lambda_0 \in [0.25, 0.5, 0.75, 0.9]$, $T_\lambda \in [20,000, 40,000, 80,000]$, and $T_\alpha \in [40,000, 80,000, 160,000]$.

## G.4 ONLINE TRAINING TIME COMPARISON

| Task | SOAR (Ours) | WSRL | PORL | Cal-QL | CQL | IQL | SAC |
|---|---|---|---|---|---|---|---|
| antmaze-large-diverse-v2 | **2.26** ± 0.01 | 2.73 ± 0.01 | 3.35 ± 0.07 | 3.44 ± 0.01 | 2.82 ± 0.02 | 2.08 ± 0.02 | 1.53 ± 0.0 |
| antmaze-large-play-v2 | 2.6 ± 0.14 | 3.48 ± 0.08 | 3.54 ± 0.06 | 3.26 ± 0.03 | **2.34** ± 0.03 | 2.09 ± 0.03 | 1.52 ± 0.0 |
| antmaze-ultra-diverse-v2 | 4.82 ± 0.12 | 4.83 ± 0.01 | 4.87 ± 0.01 | 5.09 ± 0.01 | **4.01** ± 0.01 | 3.32 ± 0.03 | 2.68 ± 0.0 |
| kitchen-partial-v0 | **3.14** ± 0.03 | 3.64 ± 0.02 | 3.78 ± 0.05 | 4.56 ± 0.03 | 3.52 ± 0.05 | 2.33 ± 0.02 | 1.71 ± 0.01 |
| kitchen-mixed-v0 | 3.62 ± 0.05 | 3.97 ± 0.05 | 3.83 ± 0.06 | 4.57 ± 0.02 | **3.6** ± 0.04 | 2.37 ± 0.05 | 1.73 ± 0.01 |
| door-binary-v0 | **1.77** ± 0.01 | 2.9 ± 0.01 | 2.93 ± 0.01 | 3.55 ± 0.01 | 2.43 ± 0.01 | 1.28 ± 0.02 | 0.78 ± 0.01 |
| pen-binary-v0 | **2.26** ± 0.01 | 2.77 ± 0.01 | 2.79 ± 0.01 | 3.51 ± 0.01 | 2.29 ± 0.01 | 1.13 ± 0.1 | 0.67 ± 0.0 |
| Average | **2.88** ± 0.17 | 3.49 ± 0.05 | 3.58 ± 0.11 | 3.91 ± 0.12 | 3.04 ± 0.14 | 2.09 ± 0.01 | 1.52 ± 0.11 |

Table 2: **Online training time. Bold** indicates the fastest training time, and underline indicates the second fastest.

Tables 3 and 4 report success rates (mean ± SE) after 400k and at 350k online fine-tuning, respectively, averaged over five seeds. Success rates for all baselines stabilize by 350k steps, validating the 350k-400k window as a reliable proxy for asymptotic performance.

SOAR achieves shorter online fine-tuning time than the baselines (Table 2). This is largely due to removing the conservative regularizer during online training, which otherwise incurs additional computational overhead. Although WSRL and PORL also drop the conservative penalty, their default high UTD ratios increase computation per environment step, resulting in longer wall-clock times than SOAR.

## G.5 REPRODUCING BASELINES

All baseline methods were reproduced under unified settings. WSRL, Cal-QL, CQL, IQL, and SAC were evaluated using the implementations provided in the WSRL codebase (Zhou et al., 2025). SOAR was also implemented on the same codebase. Since there was no public implementation of PORL, we reimplemented it following the methodology described in Xiao et al. (2025).

### G.6 EXPERIMENTAL SETUP AND REPRODUCIBILITY

All experiments were conducted on a compute node equipped with 8 NVIDIA A100 GPUs (80GB each), an AMD EPYC 7543 32-Core CPU, and 885 GB of RAM. The software environment was based on Python 3.10.18, Pytorch 2.7.0, JAX 0.4.25 with CUDA 12.2.

### G.7 PSEUDOCODE OF SOAR

---

**Algorithm 1** SOAR: Smooth Offline-to-Online Annealing for RL (concise)

---

1: **Inputs:** offline dataset $\mathcal{D}_{\text{off}}$, online steps $N_{\text{on}}$, offline steps $N_{\text{off}}$, schedules $(\lambda_0, T_\lambda)$ and $(\alpha_0, r, T_\alpha)$
2: **Initialize:** critic $Q_\theta$, actor $\pi_\phi$; online buffer $\mathcal{D}_{\text{on}} \leftarrow \emptyset$
3: **Offline pre-training (CQL)**
4: **for** $i = 1$ to $N_{\text{off}}$ **do**
5:     Sample batch $\mathcal{B} \sim \mathcal{D}_{\text{off}}$; update $Q_\theta, \pi_\phi$ with CQL using weight $\alpha_0$
6: **end for**
7: **Online fine-tuning with annealing**
8: **for** $t = 1$ to $N_{\text{on}}$ **do**
9:     Collect transition $(s_t, a_t, r_t, s_{t+1})$ with $\pi_\phi$; append to $\mathcal{D}_{\text{on}}$
10:     Compute schedules: $\lambda_t = \max\{0, \lambda_0(1 - t/T_\lambda)\}, \quad \alpha_t = \begin{cases} \alpha_0 \exp(-r\, t/T_\alpha), & t \leq T_\alpha \\ 0, & t > T_\alpha \end{cases}$
11:     Sample mixed batch $\mathcal{B}_t \sim \lambda_t\, \mathcal{D}_{\text{off}} + (1 - \lambda_t)\, \mathcal{D}_{\text{on}}$
12:     Update $Q_\theta$ and $\pi_\phi$ with the *same* CQL losses as offline, but using $\mathcal{B}_t$ and weight $\alpha_t$
13: **end for**

---

## H LIMITATIONS

While SOAR provides a simple and effective recipe for bridging offline pre-training and online fine-tuning, several limitations remain.

**Scope of evaluation.** Our experiments focus on continuous-control benchmarks (AntMaze, FrankaKitchen, Adroit) under a single-agent. We do not evaluate hard safety constraints, non-stationary dynamics, or multi-agent settings; transfer to these regimes is not guaranteed.

**Hand-designed schedules.** The linear (data ratio) and exponential (conservative weight) schedules require environment-level horizon parameters and temperature/interval choices. Although ablations indicate robustness, some sensitivity remains, and selecting schedules still needs modest tuning. We do not learn schedules adaptively or condition them on online confidence/uncertainty.

**Diagnostic constraints.** Our primary diagnostic, SQOR, is post-hoc and requires a converged critic checkpoint for comparison; this limits its use as a real-time control signal. Moreover, SQOR establishes a strong correlation with collapse but not causality; alternative mechanisms could co-vary in settings we did not test.

**Statistical coverage.** Unless otherwise noted, we use a limited number of seeds due to computational constraints. Although confidence intervals are reported, rare failure modes may be under-sampled.

## I LLM USAGE

We used a large language model (LLM) solely for language editing. Concretely, the LLM assisted with grammar and style polishing, LaTeX phrasing (e.g., equation and caption wording), and improving clarity and concision of author-written text. The LLM was not used to generate ideas, design algorithms, select hyperparameters, run experiments, analyze data, create figures/tables, write code, or produce mathematical results.

## J  VISUALIZATIONS

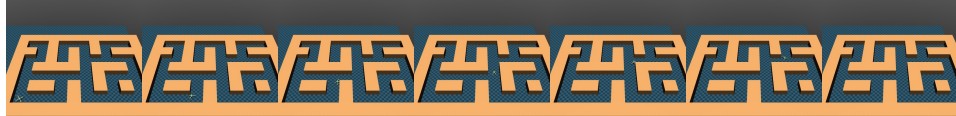

(a) **antmaze-large-diverse-v2:** 8-DoF ant navigating through a maze to reach the goal.

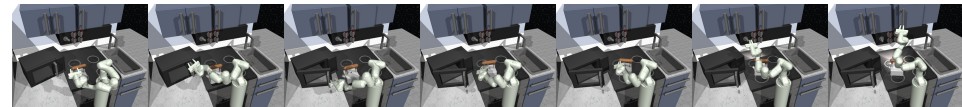

(b) **antmaze-large-play-v2:** Same evaluation as diverse-v2, different training data.

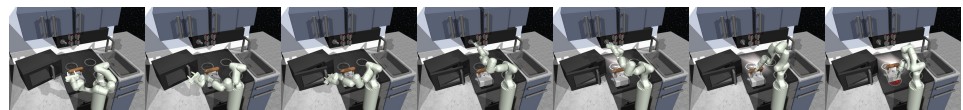

(c) **kitchen-partial-v0:** Completing subtasks: microwave, kettle, light switch, and slide cabinet.

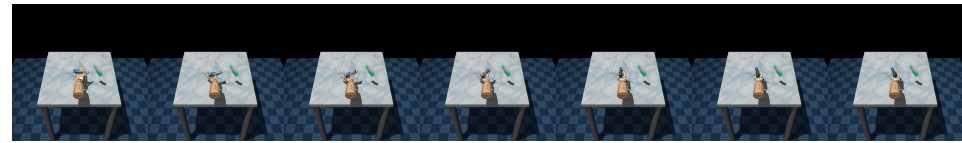

(d) **kitchen-mixed-v0:** Completing subtasks: microwave, kettle, bottom burner, and light switch.

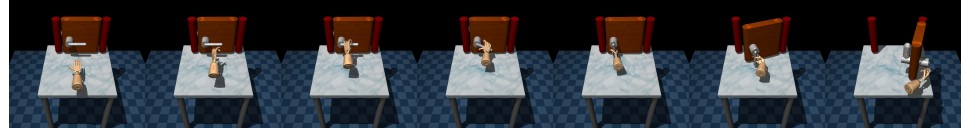

(e) **pen-binary-v0:** 24-DoF shadow hand reorienting a pen to target pose.

(f) **door-binary-v0:** 28-DoF hand opening a door by rotating the handle.

Figure 14: **Representative episodes across six tasks.** Each row shows temporally ordered frames from a trajectory, demonstrating navigation (antmaze-large-diverse-v2, antmaze-large-play-v2), kitchen manipulation (kitchen-partial-v0, kitchen-mixed-v0), and dexterous control (pen-binary-v0, door-binary-v0).

# K   NUMERICAL RESULTS

| Task | SOAR (Ours) | WSRL | PORL | Cal-QL | CQL | IQL | SAC |
|---|---|---|---|---|---|---|---|
| antmaze-large-diverse-v2 | 0.96 ± 0.02 | 0.68 ± 0.17 | 0.9 ± 0.07 | **0.97** ± 0.01 | 0.92 ± 0.05 | 0.06 ± 0.05 | 0.0 ± 0.0 |
| antmaze-large-play-v2 | 0.93 ± 0.02 | **0.95** ± 0.02 | 0.47 ± 0.2 | 0.9 ± 0.07 | **0.95** ± 0.03 | 0.01 ± 0.01 | 0.0 ± 0.0 |
| kitchen-partial-v0 | **0.93** ± 0.04 | **0.93** ± 0.06 | 0.91 ± 0.06 | 0.7 ± 0.09 | 0.44 ± 0.12 | 0.4 ± 0.05 | 0.49 ± 0.07 |
| kitchen-mixed-v0 | **0.94** ± 0.05 | 0.72 ± 0.11 | 0.88 ± 0.06 | 0.45 ± 0.12 | 0.4 ± 0.06 | 0.31 ± 0.09 | 0.3 ± 0.05 |
| door-binary-v0 | **1.0** ± 0.0 | 0.8 ± 0.2 | 0.6 ± 0.24 | 0.19 ± 0.19 | 0.18 ± 0.18 | 0.0 ± 0.0 | 0.2 ± 0.2 |
| pen-binary-v0 | 0.98 ± 0.02 | 0.97 ± 0.01 | **0.99** ± 0.01 | 0.94 ± 0.02 | 0.81 ± 0.09 | 0.33 ± 0.1 | 0.78 ± 0.04 |
| Average | **0.82** ± 0.06 | 0.72 ± 0.07 | 0.68 ± 0.07 | 0.62 ± 0.06 | 0.55 ± 0.07 | 0.16 ± 0.03 | 0.25 ± 0.06 |

Table 3: **Performance at 400k online steps. Bold** indicates the best performance per task, and underline indicates the second best.

| Task | SOAR (Ours) | WSRL | PORL | Cal-QL | CQL | IQL | SAC |
|---|---|---|---|---|---|---|---|
| antmaze-large-diverse-v2 | **0.99** ± 0.01 | 0.73 ± 0.18 | 0.84 ± 0.07 | 0.95 ± 0.03 | 0.96 ± 0.02 | 0.03 ± 0.03 | 0.0 ± 0.0 |
| antmaze-large-play-v2 | 0.9 ± 0.03 | **0.98** ± 0.01 | 0.37 ± 0.23 | 0.89 ± 0.07 | 0.95 ± 0.03 | 0.01 ± 0.01 | 0.0 ± 0.0 |
| kitchen-partial-v0 | **0.97** ± 0.02 | 0.95 ± 0.05 | 0.9 ± 0.06 | 0.7 ± 0.09 | 0.44 ± 0.12 | 0.39 ± 0.05 | 0.47 ± 0.06 |
| kitchen-mixed-v0 | **0.95** ± 0.02 | 0.7 ± 0.12 | 0.88 ± 0.06 | 0.45 ± 0.12 | 0.4 ± 0.06 | 0.3 ± 0.08 | 0.25 ± 0.03 |
| door-binary-v0 | **0.98** ± 0.01 | 0.8 ± 0.2 | 0.6 ± 0.24 | 0.17 ± 0.17 | 0.19 ± 0.19 | 0.0 ± 0.0 | 0.2 ± 0.2 |
| pen-binary-v0 | **1.0** ± 0.0 | 0.96 ± 0.02 | 0.95 ± 0.02 | 0.96 ± 0.01 | 0.72 ± 0.05 | 0.4 ± 0.05 | 0.84 ± 0.02 |
| Average | **0.84** ± 0.06 | 0.72 ± 0.07 | 0.65 ± 0.07 | 0.63 ± 0.06 | 0.54 ± 0.06 | 0.16 ± 0.03 | 0.25 ± 0.06 |

Table 4: **Performance at 350k online steps. Bold** indicates the best performance per task, and underline indicates the second best.

Tables 3 and 4 report success rates (mean ± SE) after 400k and at 350k online fine-tuning, respectively, averaged over five seeds. Success rates for all baselines stabilize by 350k steps, validating the 350k-400k window as a reliable proxy for asymptotic performance. Offline-to-online methods consistently outperform offline-only approaches, and SOAR achieves the highest average success rate.

| Task | SOAR (Ours) | WSRL | PORL | Cal-QL | CQL | IQL |
|---|---|---|---|---|---|---|
| antmaze-large-diverse-v2 | 0.34 ± 0.04 | 0.24 ± 0.05 | 0.39 ± 0.04 | 0.26 ± 0.04 | 0.26 ± 0.05 | 0.03 ± 0.02 |
| antmaze-large-play-v2 | 0.28 ± 0.04 | 0.29 ± 0.02 | 0.37 ± 0.02 | 0.24 ± 0.04 | 0.28 ± 0.05 | 0.0 ± 0.0 |
| kitchen-partial-v0 | 0.68 ± 0.06 | 0.73 ± 0.06 | 0.7 ± 0.04 | 0.68 ± 0.05 | 0.7 ± 0.06 | 0.3 ± 0.06 |
| kitchen-mixed-v0 | 0.47 ± 0.06 | 0.47 ± 0.06 | 0.49 ± 0.07 | 0.52 ± 0.06 | 0.58 ± 0.03 | 0.4 ± 0.04 |
| door-binary-v0 | 0.2 ± 0.05 | 0.31 ± 0.19 | 0.24 ± 0.03 | 0.21 ± 0.07 | 0.08 ± 0.08 | 0.05 ± 0.04 |
| pen-binary-v0 | 0.73 ± 0.07 | 0.66 ± 0.05 | 0.74 ± 0.03 | 0.75 ± 0.05 | 0.58 ± 0.03 | 0.89 ± 0.04 |
| Average | 0.42 ± 0.04 | 0.41 ± 0.04 | 0.45 ± 0.04 | 0.4 ± 0.04 | 0.39 ± 0.04 | 0.25 ± 0.05 |

Table 5: **Performance after the offline phase.**

Table 5 reports success rates (mean ± SE) at the end of the offline pre-training phase, i.e., the start of online fine-tuning, averaged over five seeds. Despite identical offline training procedures, end-of-offline performance shows small variations across SOAR, WSRL, PORL, Cal-QL, and CQL. These differences reflect evaluation stochasticity rather than algorithmic effects.

| Task | SOAR (Ours) | WSRL | PORL | Cal-QL | CQL | IQL |
|---|---|---|---|---|---|---|
| antmaze-large-diverse-v2 | $\underline{0.15} \pm 0.05$ | $\mathbf{0.07} \pm 0.04$ | $0.39 \pm 0.04$ | $0.17 \pm 0.07$ | $0.21 \pm 0.04$ | $0.03 \pm 0.02$ |
| antmaze-large-play-v2 | $\mathbf{0.08} \pm 0.03$ | $0.2 \pm 0.06$ | $0.37 \pm 0.02$ | $\underline{0.16} \pm 0.05$ | $\underline{0.16} \pm 0.07$ | $0.0 \pm 0.0$ |
| kitchen-partial-v0 | $\mathbf{0.6} \pm 0.08$ | $\underline{0.68} \pm 0.09$ | $0.7 \pm 0.04$ | $\underline{0.68} \pm 0.05$ | $0.7 \pm 0.06$ | $0.19 \pm 0.04$ |
| kitchen-mixed-v0 | $\mathbf{0.36} \pm 0.1$ | $\underline{0.42} \pm 0.09$ | $0.48 \pm 0.07$ | $0.52 \pm 0.06$ | $0.58 \pm 0.03$ | $0.35 \pm 0.04$ |
| door-binary-v0 | $\mathbf{0.2} \pm 0.05$ | $0.31 \pm 0.19$ | $0.24 \pm 0.03$ | $\underline{0.21} \pm 0.07$ | $0.08 \pm 0.08$ | $0.05 \pm 0.04$ |
| pen-binary-v0 | $\mathbf{0.32} \pm 0.08$ | $0.52 \pm 0.04$ | $0.66 \pm 0.04$ | $\underline{0.33} \pm 0.08$ | $0.58 \pm 0.03$ | $0.87 \pm 0.03$ |
| Average | $\mathbf{0.28} \pm 0.04$ | $0.34 \pm 0.05$ | $0.44 \pm 0.03$ | $\underline{0.31} \pm 0.04$ | $0.36 \pm 0.04$ | $0.23 \pm 0.05$ |

Table 6: **Catastrophic failure (0-400K window). Bold** indicates the lowest catastrophic failure, and underline indicates the second lowest.

| Task | SOAR (Ours) | WSRL | PORL | Cal-QL | CQL | IQL |
|---|---|---|---|---|---|---|
| antmaze-large-diverse-v2 | $\underline{0.15} \pm 0.05$ | $\mathbf{0.07} \pm 0.04$ | $0.39 \pm 0.04$ | $0.17 \pm 0.07$ | $0.21 \pm 0.04$ | $0.03 \pm 0.02$ |
| antmaze-large-play-v2 | $\mathbf{0.08} \pm 0.03$ | $0.2 \pm 0.06$ | $0.37 \pm 0.02$ | $\underline{0.16} \pm 0.05$ | $\underline{0.16} \pm 0.07$ | $0.0 \pm 0.0$ |
| kitchen-partial-v0 | $\mathbf{0.55} \pm 0.09$ | $\underline{0.68} \pm 0.09$ | $0.7 \pm 0.04$ | $\underline{0.68} \pm 0.05$ | $0.7 \pm 0.06$ | $0.18 \pm 0.04$ |
| kitchen-mixed-v0 | $\mathbf{0.32} \pm 0.1$ | $\underline{0.42} \pm 0.09$ | $0.48 \pm 0.07$ | $0.52 \pm 0.06$ | $0.58 \pm 0.03$ | $0.33 \pm 0.03$ |
| door-binary-v0 | $\mathbf{0.2} \pm 0.05$ | $0.31 \pm 0.19$ | $0.24 \pm 0.03$ | $\underline{0.21} \pm 0.07$ | $0.08 \pm 0.08$ | $0.05 \pm 0.04$ |
| pen-binary-v0 | $\mathbf{0.32} \pm 0.08$ | $0.52 \pm 0.04$ | $0.66 \pm 0.04$ | $\underline{0.33} \pm 0.08$ | $0.58 \pm 0.03$ | $0.83 \pm 0.04$ |
| Average | $\mathbf{0.26} \pm 0.03$ | $0.34 \pm 0.05$ | $0.44 \pm 0.03$ | $\underline{0.31} \pm 0.04$ | $0.36 \pm 0.04$ | $0.22 \pm 0.05$ |

Table 7: **Catastrophic failure (0-100K window). Bold** indicates the lowest catastrophic failure, and underline indicates the second lowest.

Tables 6 and 7 report catastrophic failure measured over the 0-400k and 0-100k windows, respectively. All values are averaged over five seeds and reported as mean $\pm$ SE. Estimates do not differ significantly between the two windows, indicating that failures arise primarily during the offline-to-online transition. SOAR attains the lowest catastrophic failure on average. For clarity, we do not highlight IQL, whose low failure largely reflects weak performance after the offline phase rather than stable transitions.

