# OpenReview forum: "Annealing Bridges Offline and Online RL"
_ICLR.cc/2026/Conference — ICLR 2026 Conference Withdrawn Submission_

### Official Review · Reviewer_E7Hj · 2025-10-21

**Soundness:** 2
**Presentation:** 2
**Contribution:** 3
**Rating:** 2
**Confidence:** 4

**Summary:**

The paper identifies squrious Q optimism as the cause of catastrophic failure during the initial phase of online fine-tuning in offline-to-online reinforcement learning.
To address this issue, the paper proposes a new algorithm called SOAR.
SOAR employs a simple annealing method to gradually reduce the reliance on offline data and decrease the strengths of the conservative penalty during the online fine-tuning.
By doing so, SOAR successfully prevents catastrophic failure and demonstrates improved asymptotic performance.

**Strengths:**

* This paper addresses catastrophic failure, one of the critical issues that needs to be tackled in offline-to-online RL. It proposes spurious Q-optimism as a new cause for this phenomenon.
* The experimental results are sufficiently provided.

**Weaknesses:**

* The writing needs significant improvement.
  * The introduction covers too many details. Clarity would be improved by making it more concise.
  * The lack of a Background or Preliminaries section will make it difficult for readers unfamilar with the field to understand the paper.
  * The overall section structure of the paper is disorganized, which significantly hinders comprehension
  * $\mathcal{L}_\text{TD}$, which appears in all equations, is not defined.
  * The paper's narrative and flow make it difficult to clearly follow the main arguments and intended message.
* Although the paper experimentally shows that SOAR is a superior algorithm, the results indicate that SOAR's performance lies within the confidence interval of other baselines in almost all environments.
* The proposed algorithm introduces a large number of hyperparameters, and the need for domain-specific tuning for each environment is a concern regarding its general applicability.
* The relationship etween SQOR and the success rate is not clear. In Antmaze (Figure 4), the SQOR values are similar, but the collapse rates differ. Conversely, in kitchen-mixed, the SQOR difference is large, but the degree of collapse is similar. A similar pattern is observed between kitchen-mixed and pen-binary in Figure 7.

**Questions:**

* In Section 4, does 'when baselines are fine-tuned without retaining offline data' mean that all results in Figure 2 are from an online fine-tuning setting without offline data, or does it only refer to specific algorithms like WSRL or SAC ?
* What is the basis or evidence for the claim in Section 5 that it 'hinder effective exploration of the optimal policy' ?
* In Section 6.1, is it necessary to define $Q_\text{final}$ at 400k online steps ? Wouldn't the value function obtained after further training be considered more accurate ?
* In Section 6.1, what action is input to the value function used in the KL divergence calculation ?
* I am unsure if SQOR is an appropriate metric from a value ordering perspective. What are the results when compared with the Kendall coefficient shown in previous study [1] ?
* There are existing study [2] that reduce conservativeness during online fine-tuning while modifying the buffer. What is the contribution of this study when compared to this method ?

&nbsp;

[1] Zhang, Y., Liu, J., Li, C., Niu, Y., Yang, Y., Liu, Y., & Ouyang, W. (2024, March). A perspective of q-value estimation on offline-to-online reinforcement learning. In Proceedings of the AAAI conference on artificial intelligence (Vol. 38, No. 15, pp. 16908-16916).

[2] Beeson, A., & Montana, G. Improving TD3-BC: Relaxed Policy Constraint for Offline Learning and Stable Online Fine-Tuning. In 3rd Offline RL Workshop: Offline RL as a''Launchpad''.

---

### Official Review · Reviewer_NPHo · 2025-10-31

**Soundness:** 3
**Presentation:** 3
**Contribution:** 3
**Rating:** 6
**Confidence:** 3

**Summary:**

This paper addresses the critical challenge in offline-to-online reinforcement learning: the tension between catastrophic failure (sharp early performance collapse during the transition from offline pre-training to online fine-tuning) and asymptotic success rates (long-term performance after convergence). The authors conduct a systematic empirical study across D4RL, Adroit, and FrankaKitchen environments, demonstrating that existing offline and offline-to-online RL methods fail to simultaneously prevent catastrophic failure and achieve high asymptotic success rates. Through careful analysis, they identify spurious Q-optimism as the primary driver of catastrophic failure, a phenomenon where the learned value function incorrectly reverses the relative ranking of actions early in fine-tuning, causing the agent to favor actions that ultimately under perform. They quantify this via a new metric called Spurious Q-Optimism Ratio (SQOR), which exhibits strong correlation with collapse across tasks. Finally, they propose SOAR (Smooth Offline-to-Online Annealing for RL), a simple dual annealing scheme that gradually reduces both the offline data fraction and the conservative regularizer weight during online fine-tuning. Extensive experiments show that SOAR reduces catastrophic failure while achieving superior asymptotic performance compared to strong baselines including WSRL, PORL, Cal-QL, CQL, and IQL

**Strengths:**

**Systematic and rigorous problem characterization.** The paper provides the most comprehensive empirical analysis of the offline-to-online transition failure modes to date. The systematic study demonstrates that existing methods fundamentally cannot balance catastrophic failure suppression with high asymptotic success, which is well-illustrated in Figure 2. The clear problem formulation and distinction between failure at onset versus long-term performance is methodologically sound and important for the field.

**Novel and actionable diagnosis of failure mechanism.** The identification of spurious Q-optimism as the primary driver of collapse is valuable, and the quantification via SQOR provides a measurable diagnostic tool. Figure 4 convincingly demonstrates that SQOR tracks catastrophic failure across diverse tasks and stress settings far better than alternative metrics (SQOG, O-SQOR, volatility). This diagnostic contribution is independently useful for understanding offline-to-online RL failure modes.

**Comprehensive ablation and robustness analysis.** The ablation studies in Figures 8, 11, and 12, as well as appendices B and C, are thorough and well-executed. The paper provides concrete guidance on single-component annealing strategies for practitioners prioritizing safety or speed, and demonstrates robustness across hyperparameter variations.

**Strong empirical results across diverse domains.** The experiments span multiple challenging continuous control environments (AntMaze, FrankaKitchen, Adroit) with both standard and ultra-diverse variants. SOAR achieves the highest average success rates (Table 3: 0.82 ± 0.06) and lowest catastrophic failure (Table 6: 0.28 ± 0.04).

**Weaknesses:**

**Limited theoretical understanding of spurious Q-optimism.** While the paper identifies spurious Q-optimism and quantifies it empirically, it lacks theoretical analysis connecting SQOR to convergence guarantees or failure probability bounds. Why does spurious Q-optimism occur? What conditions make it more or less likely? The paper would be significantly strengthened by formal analysis of how distributional shift, offline dataset properties, and critic initialization interact to create value ordering reversals.

**Missing comparisons with recent related work.** The paper does not compare with some relevant recent offline-to-online and safe RL methods that address similar concerns (e.g., uncertainty-guided methods, policy distillation approaches, or implicit value regularization variants). Including these comparisons would better position SOAR within the current landscape.

**Insufficient analysis of computational overhead.** While Table 2 reports online training time, the paper does not provide detailed wall-clock time breakdowns or discuss the computational cost of computing SQOR diagnostics if practitioners wish to monitor collapse in real time. The comparison to WSRL and PORL shows SOAR is competitive, but more analysis would be helpful.

**Questions:**

**Q1:** Can you provide theoretical or empirical justification for the specific annealing schedules? Why use linear decay for offline data ratio but exponential decay for the conservative weight (Equation 2)? Can you derive or justify these choices from first principles?

**Q2:** How sensitive is SOAR to the offline pre-training quality and dataset composition? The paper uses standard D4RL datasets. How does SOAR perform when offline data is of poor quality (i.e, random), or significantly mismatched to the online task? Does the method degrade?

**Q8:** What is the theoretical sample complexity of SOAR? Does the annealing schedule affect the sample efficiency of online learning, or are there regret bounds that characterize SOAR's performance?

---

### Official Review · Reviewer_kWGK · 2025-10-31

**Soundness:** 2
**Presentation:** 3
**Contribution:** 2
**Rating:** 4
**Confidence:** 3

**Summary:**

This work investigates the offline-to-online reinforcement learning (O2O RL) through analyses of catastrophic performance degradation during the early fine-tuning phase. They identify Spurious Q-Optimism Ratio (SQOR), which is a metric over erroneous Q-value ranking between offline and online actions, as the principal cause of performance degradation.
Building on the analyses, they propose Smooth Offline-to-Online Annealing for RL (SOAR), a CQL-based fine-tuning scheme that uses dual annealing of two factors: the conservative regularization weight and the offline/online data ratio. By gradually decaying both, SOAR aims to mitigate catastrophic degradation while maintaining strong asymptotic performance.

**Strengths:**

S1. (Simple method)
The proposed dual annealing strategy is algorithmically minimal, requiring only a time-varying schedule for offline data ratio and Q-value regularization coefficient without introducing additional components.

S2. (Clear motivation)
Based on their analyses, the dual annealing strategy is well-motivated.

S3. (Extensive analyses)
Empirical analyses about CQL and ablation studies about SOAR support the main claim that both components of the annealing schedule contribute meaningfully.

**Weaknesses:**

W1. (Missing evaluation under offline-data retention)
Figure 2 presents results only for settings without retaining offline data. However, since the core contribution of SOAR involves annealing from an initial offline mixture, a comparison with offline data retention would be necessary for a fair evaluation. Unlike WSRL, SOAR does not advocate for the complete removal of offline data, therefore, both cases should be reported for more comprehensive assessment.

W2. (Limited generalizability)
The proposed dual annealing strategy is specifically implemented within the CQL framework for online fine-tuning. It remains unclear whether the strategy can be effectively integrated into other CQL-style or conservative offline-to-online RL methods. This uncertainty limits the general applicability of SOAR as a broadly usable O2O RL method.

W3. (Unclear strength and interpretability of SQOR)
The correlation between SQOR and performance degradation is not convincingly demonstrated in Figure 4. Moreover, since SQOR evaluates only states drawn from the offline dataset, its validity is limited under large state-distribution shifts that naturally occur during online fine-tuning. Consequently, it is uncertain whether SQOR serves as an appropriate metric for analyzing online adaptation behavior. In addition, it remains unclear whether SQOR is meaningful for other offline or offline-to-online algorithms beyond CQL.

W4. (Limited practical utility of SQOR)
SQOR requires access to the final critic to compute the sign mismatch between current and converged Q-values, which constrains its practical use during online fine-tuning. If a method were proposed to incorporate SQOR estimation during online fine-tuning, it could potentially enhance the practical impact of the work.

**Minor**
- Duplicated sentences at lines 63 to 66.

**Questions:**

Could you provide additional analyses or discussions addressing these weaknesses?

---

### Official Review · Reviewer_aefe · 2025-11-01

**Soundness:** 1
**Presentation:** 2
**Contribution:** 1
**Rating:** 2
**Confidence:** 5

**Summary:**

This paper studies the trade off between avoiding catastrophic failure and asymptotic performance when moving to the online fine-tuning phase in offline to online RL. It performs analysis, and proposes a small modification on top of the standard fine-tuning recipe.

**Strengths:**

- SQOR is an interesting metric to keep track of to understand stability of offline-to-online RL algorithms.
- Some analysis was provided to understand what matters in preventing catastrophic failure.

**Weaknesses:**

This paper significantly misrepresents most of its key results. Further, the method itself is a trivial modification of the standard online RL fine-tuning recipe.

- Lines 63-66 have a repeated sentence.
- Explanation of SQOR (line 71) is unclear - what does “current versus final value ordering” mean?
- “we extend finetuning steps to 400K steps to better observe asymptotic performance trends.” The difference between 300k and 400k is so small that mentioning this takes away focus from other important things.
- “On Adroit, we found that increasing pen-binary pre-training from 20K (used in WSRL) to 40K yields more consistent gains; for door-binary, offline pre-training variance is higher and the difference between 20K and 40K is less pronounced, but we adopt 40K to stabilize trends.” Did you do this for all baselines? What does “stabilize trends” mean? Again, these details take focus out of the important parts of your paper.
- “For PORL, we adopt the authors’ settings described in Xiao et al. (2025).” Did you also try the modifications you mention in G.3 for PORL?
- “This motivates a controlled analysis of how removing the conservative regularizer affects both outcomes” This sentence promises analysis, but immediately after, section 4 is finished, and no analysis is in sight.
- “the availability of offline data is a key distinction between the offline and online phase” This sentence is unclear. During both phases offline data is available, but the algorithm may choose not to use it.
- Figure 3: What does O/X stand for? E.g. “offline data O”
- Figure 3 a) “keeping conservative regularizer reduces early collapses” in 2/4 environments early collapses are not statistically significantly reduced.
- Figure 3 b) “Keeping α lowers asymptotic performance.” My understanding of row b is that it is comparing with and without conservative regularizer but without offline data (offline data X). 3 out of 4 plots in this row show performance that is not statistically significantly different between the 2 methods.
- “In contrast, annealing the offline fraction to zero, as in our method (Section 7.1), yields faster convergence and final performance comparable to retaining offline data” Figure 3 (d) absolutely does not show faster convergence for the offline data annealing.
- Overall, the analysis of Figure 3 is severely misrepresenting the facts.
- Figure 2: For most environments, all methods are compressed on the upper bound of performance. The differences between methods mainly come from door-binary-v0. The suite used in this paper is simply not useful to disambiguate between methods.
- The authors included only some environments from the environmental suites they considered. E.g., they skipped kitchen-complete-v0, relocate-binary-v0. This is a red flag.
- “SOAR consistently reduces catastrophic failure and improves asymptotic performance across tasks (Figure 2)” SOAR still shows massive catastrophic failures in Figure 2, and does not improve asymptotic performance in most tasks.
- Code is not released.

**Questions:**

- Why was kitchen-complete-v0 omitted?
- Why was CQL used for pre-training instead of Cal-QL, since Cal-QL is better suited for online fine-tuning?

---

### Note · Authors · 2025-11-26

**Comment:**

We have decided to withdraw the submission in order to further strengthen both the theoretical foundations and empirical evidence. We believe additional analysis and experiments will allow us to present a more complete and rigorous version of the work in a future submission.

**Withdrawal Confirmation:**

I have read and agree with the venue's withdrawal policy on behalf of myself and my co-authors.